# DEXOM: Diversity-based enumeration of optimal context-specific metabolic networks

**Pablo Rodríguez-Mier**[1], **Nathalie Poupin**[1], **Carlo de Blasio**[2,3], **Laurent Le Cam**[2,3], **Fabien Jourdan**[1]*

**1** Toxalim (Research Centre in Food Toxicology), Université de Toulouse, INRAE, ENVT, INP-Purpan, UPS, Toulouse, France, **2** IRCM, Institut de Recherche en Cancérologie de Montpellier, INSERM U1194, Université de Montpellier, Institut régional du Cancer de Montpellier, Montpellier, France, **3** Equipe Labellisée par la Ligue contre le Cancer, Paris, France

* fabien.jourdan@inrae.fr

**Data Availability Statement:** All data and methods are available from https://github.com/MetExplore/dexom.

## Abstract

The correct identification of metabolic activity in tissues or cells under different conditions can be extremely elusive due to mechanisms such as post-transcriptional modification of enzymes or different rates in protein degradation, making difficult to perform predictions on the basis of gene expression alone. Context-specific metabolic network reconstruction can overcome some of these limitations by leveraging the integration of multi-omics data into genome-scale metabolic networks (GSMN). Using the experimental information, context-specific models are reconstructed by extracting from the generic GSMN the sub-network most consistent with the data, subject to biochemical constraints. One advantage is that these context-specific models have more predictive power since they are tailored to the specific tissue, cell or condition, containing only the reactions predicted to be active in such context. However, an important limitation is that there are usually many different sub-networks that optimally fit the experimental data. This set of optimal networks represent alternative explanations of the possible metabolic state. Ignoring the set of possible solutions reduces the ability to obtain relevant information about the metabolism and may bias the interpretation of the true metabolic states. In this work we formalize the problem of enumerating optimal metabolic networks and we introduce DEXOM, an unified approach for diversity-based enumeration of context-specific metabolic networks. We developed different strategies for this purpose and we performed an exhaustive analysis using simulated and real data. In order to analyze the extent to which these results are biologically meaningful, we used the alternative solutions obtained with the different methods to measure: 1) the improvement of in silico predictions of essential genes in *Saccharomyces cerevisiae* using ensembles of metabolic network; and 2) the detection of alternative enriched pathways in different human cancer cell lines. We also provide DEXOM as an open-source library compatible with COBRA Toolbox 3.0, available at https://github.com/MetExplore/dexom.

**Funding:** This work was supported by grants from the Institut National contre le Cancer (INCa-PLBIO) and the Labex EpiGenMed ("Investissements d'avenir" program, reference ANR-10-LABX-12-01). This work has also received funding from the European Union's Horizon 2020 research and innovation program under grant agreement GOLIATH No. 825489, and from the Association pour la Recherche contre le Cancer (ARC) Foundation. The funders had no role in study design, data collection and analysis, decision to publish, or preparation of the manuscript.

**Competing interests:** The authors have declared that no competing interests exist.

## Author summary

Understanding deregulations of metabolism based on isolated measures of gene expression or protein or metabolite concentrations is a challenging task due to the interconnection of multiple processes. One solution is to extract, from generic genome-scale metabolic networks, the specific sub-network which is modulated in the studied condition. Many algorithms have been proposed for such context-specific network extraction based on experimental measurements. However, this process is subject to some randomness and variability, since multiple metabolic networks can model the metabolic state in a similarly adequate manner for the same experimental data. This means that for a given data and reconstruction method, there are usually multiple solutions that satisfy the same constraints and with the same quality, but only one solution is returned by the commonly used reconstruction methods. Here, we formalize this problem and we propose and analyze different methods to obtain diverse samples of metabolic sub-networks. We evaluate them by performing an extensive comparison and we show how the different sets of optimal networks discovered by the different methods are biological meaningful by constructing ensembles of networks to improve the prediction of essential genes in *Saccharomyces cerevisiae* and to detect enriched metabolic pathways in four different human cancer cell lines.

This is a *PLOS Computational Biology* Methods paper.

## Introduction

Metabolism and its regulation is an ensemble of intricate and tightly coordinated processes involving hundreds to thousands of enzymes, reactions, metabolites and genes, whose interactions define complex networks that are unique for each species. This complexity grants organisms the flexibility to adapt their energetic functions and growth requirements to a wide variety of conditions. Changes in nutrient availability, conditions of cellular stress, or any other change in the environment can induce a rapid metabolic reprogramming of cells, rewiring their metabolism to adjust to the requirements of the new situation. Dysfunction of these mechanisms play a central role in the development of many diseases, but most notably in cancer, where cancer cells exploit metabolic reprogramming on their own benefit [1] to sustain a rapid proliferation rate and survive in conditions of hypoxia, nutrient depletion, or even develop therapy resistance [2]. Being able to accurately detect these changes or deregulations in metabolism would be beneficial not only for a better understanding of biological systems but to develop more targeted therapies and treatments for many diseases [3–5].

One of the reasons why this task remains elusive is the complexity of the multiple processes that participate in the regulation of the metabolism [6]. More specifically, post-transcriptional control of mRNA, post-translational modifications of enzymes, as well as biochemical constraints —including for example the laws for mass and charge conservation, cell growth requirements, biomass composition and nutrient availability— make the identification of which pathways are altered between conditions very complicated by the mere observation of changes in gene expression or changes in metabolite concentrations. Instead, integrating and analyzing together all those different levels of information is key to improve the predictive models and to provide a more accurate mechanistic view of the system under study.

Genome-scale metabolic networks (GSMN) are suitable computational models for the integration of these multiple levels of knowledge. These models are automatically built and

manually curated networks that encode all reactions with their stoichiometric coefficients, metabolites, enzymes, gene annotations and biochemical constraints that are known for an organism. GSMNs are generic models of an organism, independent of the type of tissue, cell or condition. In order to generate more accurate models for specific tissues or conditions, experimental data such as gene or protein expression can be integrated on top of GSMNs using context-specific network reconstruction methods. Taking into account the different levels of expression of genes between conditions, a sub-network from the GSMN is extracted by finding a steady-state flux most consistent with the experimental data. This process allows the generation of metabolic networks specifically tailored to the condition, to highlight for example differences in metabolism between tissues [7–9] or to discover novel drug targets or essential genes in cancer cells [10–12].

Several methods were proposed in the literature to automatically reconstruct context-specific metabolic networks from gene or protein expression, mostly based on Linear Programming (LP) or Mixed Integer Linear Programming (MILP) models [7–9, 13–17], as well as benchmarks comparing their capabilities [18, 19]. This process is done by solving an optimization problem to find the sub-network from the GSMN that maximizes the agreement with the experimental data. This agreement is defined in different ways: some methods such as [7, 15] use data to classify reactions into reactions associated to highly expressed enzymes (or core reactions) or reactions associated to lowly expressed enzymes, whereas others [8, 14] assign different scores (weights) to reactions based on data and other experimental evidence. The optimization problem is then defined as that of finding the sub-networks that can carry a steady-state flux through the reactions that maximize the overall score. However, a major limitation that is frequently neglected is that the available information is usually not sufficient to fully and unambiguously characterize the corresponding metabolic sub-network for a given condition. Instead, a range of different optimal metabolic sub-networks may exist, offering different hypotheses of the possible metabolic state. In other words, for a given experimental data, reconstruction method, and pre-processing method to score the importance of the reactions (e.g., threshold-based methods to classify reactions into active or non active), there exist an unknown amount of possible metabolic sub-networks (solutions) that are equally valid (optimal) in terms of agreement with experimental data, but only one of these solutions (without knowing which one a priori) is returned by the commonly used context-specific reconstruction methods. Ignoring this variability can not only lead to incorrect or incomplete explanations of the biological experiment, but also causes valuable information to be lost that could be used to improve predictions. Although this limitation is starting to be acknowledged [20–22], there is still a lack of studies that analyze the computational problem and that provide methods to sample or enumerate the optimal space of alternative networks.

The problem of exploring multiple solutions in the context of metabolic networks was already carried out for Flux Balance Analysis [23], but barely analyzed for context-specific network reconstruction, where both the type of the problem and purpose are different. One of the initial works that exploits the idea of multiple context-specific networks to improve predictions is EXAMO [21]. In this work, authors perform an enumeration of optimal metabolic networks using iMAT [7]. The enumeration is done using the same strategy proposed in iMAT for assigning confidence scores to reactions, followed by a post-processing step using the MBA [13] algorithm to generate a single consensus network including the reactions predicted to be active. A similar strategy was applied by Poupin et al. [20], but instead of generating a single consensus network, the whole set of networks derived by forcing fluxes through each reaction in the model is preserved as alternative hypotheses of the metabolic state. This enables a better characterization of the metabolic shifts that occur during hepatic differentiation.

The procedure of generating alternative networks by forcing or blocking flux through each reaction has however some limitations. First, it can generate many duplicated solutions. For example, if there exist only one optimal metabolic network with a linear pathway of 10 reactions, forcing the activation of each reaction in the linear pathway will generate always the same optimal solution, wasting computational resources. Second, it cannot recover the whole set of possible optimal metabolic networks, as not all possible combinations of reactions are tested. Third, there is no guarantee that the solution set is representative and diverse of the full space of possible networks. A simple brute force algorithm that could be used to prevent this would be to test every possible combination between variable reactions. However, this approach does not scale as the number of problems to solve grows exponentially with the number of variable reactions. As an alternative to this approach, authors in [22] present a strategy to generate alternative metabolic networks. Of particular interest is their `CorEx` algorithm, which in a similar fashion as `Fastcore` method [15], calculates the smallest flux-consistent sub-network that preserve the reactions in the core set, but solving the problem exactly instead of the LP-based fast approximations used in `Fastcore`. `CorEx` also incorporates a mechanism to enumerate optimal networks by maximizing the dissimilarity with the previously found solution, a process that can be repeated iteratively to discover new optimal networks. However, without a mechanism that prevents the generation of duplicated solutions, the enumeration process can get stuck in a small region in the space of optimal solutions. Some issues still remain with this enumeration strategy, mainly regarding its effectiveness to get a representative set of the possible metabolic networks and also how to take advantage of the set of networks to improve predictions more than just only observing the variability in terms of reactions that can appear or not in the different optimal sub-networks.

Regarding this last question, it was shown that the use of ensembles of draft networks reconstructed using Gap Filling methods with multiple media conditions and random perturbations can improve flux-based predictions [24]. Although the application is different, predictions using context-specific network reconstruction methods could be also improved using ensembles of optimal metabolic networks, and diversity can play an important role in the quality of the ensemble models.

In this work, we advocate for generating a diverse set of solutions, that is, given some experimental condition for which we cannot characterize the metabolic state with just one optimal network, we want to obtain a sample of this largely unknown set of possible networks in a way that covers well the range of possibilities. In other words: if large differences in metabolism can be explained by the same experimental data, we want to obtain a diverse set of these optimal networks that capture those different metabolic states. This usually means exploring distant solutions with changes that correspond also to distant pathways.

The concept of diversity of optimal solutions of a MILP problem is not well explored in metabolic network reconstruction, and only marginally analyzed in combinatorial optimization. Of special interest is the sequential MILP approach proposed by Danna et al. [25], in which they propose an enumeration strategy which incorporates the concept of diversity by maximizing the distance to previously found solutions at the same time that they discard visited solutions. The closest concept to this general strategy applied to the enumeration of optimal context-specific metabolic networks can be found in [22], where Robaina et al. incorporate the idea of maximizing the distance to the previous solution, but without a mechanism that would remove already explored solutions.

Although maximizing the distance may seem like a good idea a priori, in practice it can lead to oscillations in the search, in which the search process jumps between two distant clusters of possible networks, with large inter-cluster distance but very small intra-cluster

distance. That is why the concept of diversity in metabolic networks must be carefully analyzed with synthetic and real data that allow observing the behavior and quality of the solutions.

As a response to the current limitations, here we formalize the problem of enumerating optimal context-specific metabolic networks from a computational perspective and we develop `DEXOM`, a collection of MILP-based methods for diversity-based enumeration of optimal metabolic networks. We implemented in total four different techniques in `DEXOM`, namely: `Reaction-enum`, `Icut-enum`, `Maxdist-enum` and `Diversity-enum`.

The objective of this paper is threefold. First, to analyze and formalize, from a computational perspective, the enumeration problem of optimal solutions and propose different practical techniques that can be used to obtain not one but many, equally good and diverse context-specific metabolic networks. Second, to analyze how the different methods behave under different simulated and real conditions and identify real examples where the discovery of a more diverse set of possible metabolic networks have practical implications. Third, to provide an unified open source library with the different implementations that can be used in a general way to find diverse solutions.

In order to evaluate how well each method performs, we focused on two main aspects: 1) how well each technique is able to discover a diverse set of optimal networks, measured using different distance metrics, and 2) how the set of alternative optimal solutions is biological meaningful by assessing the predictive capabilities with real data. We performed in total around 191,000 network reconstructions with simulated data, 329,000 using microarray data from *Saccharomyces cerevisiae* [18, 26] and around 67,400 using RNA-seq data from different human cancer cell lines [19]. To analyze the extent to which these results are biologically meaningful, we used these reconstructions to measure: 1) the improvement of in silico predictions of essential genes in yeast using ensembles of metabolic network; and 2) the detection of alternative enriched pathways in human cancer cells, as a way to measure the variability of different hypotheses about the metabolic state that are compatible with the experimental data (Fig 1).

To summarize, the main novelties of this work are:

- The analysis and identification of the computational problem involving the diversity-based enumeration of optimal context-specific metabolic network.

- The development of a library (`DEXOM`) including four different methods (`Reaction-enum`, `Icut-enum`, `Maxdist-enum` and `Diversity-enum`) for the enumeration of optimal context-specific metabolic networks.

- An extensive comparison using the different methods under different experimental conditions, showing how variable the spacing of valid optimal solutions usually is, and comparing the methods in terms of ability to detect these solutions.

- The development of an open-source library integrated with COBRA Toolbox 3.0.6 [27] with the different methods for the enumeration of solutions, available at https://github.com/MetExplore/dexom

## Methods

In this section we introduce the problem of context specific metabolic network reconstruction and the enumeration problem, we describe the four different strategies that we implemented in `DEXOM` namely: `Reaction-enum`, `Icut-enum`, `Maxdist-enum` and `Diversity-enum`.

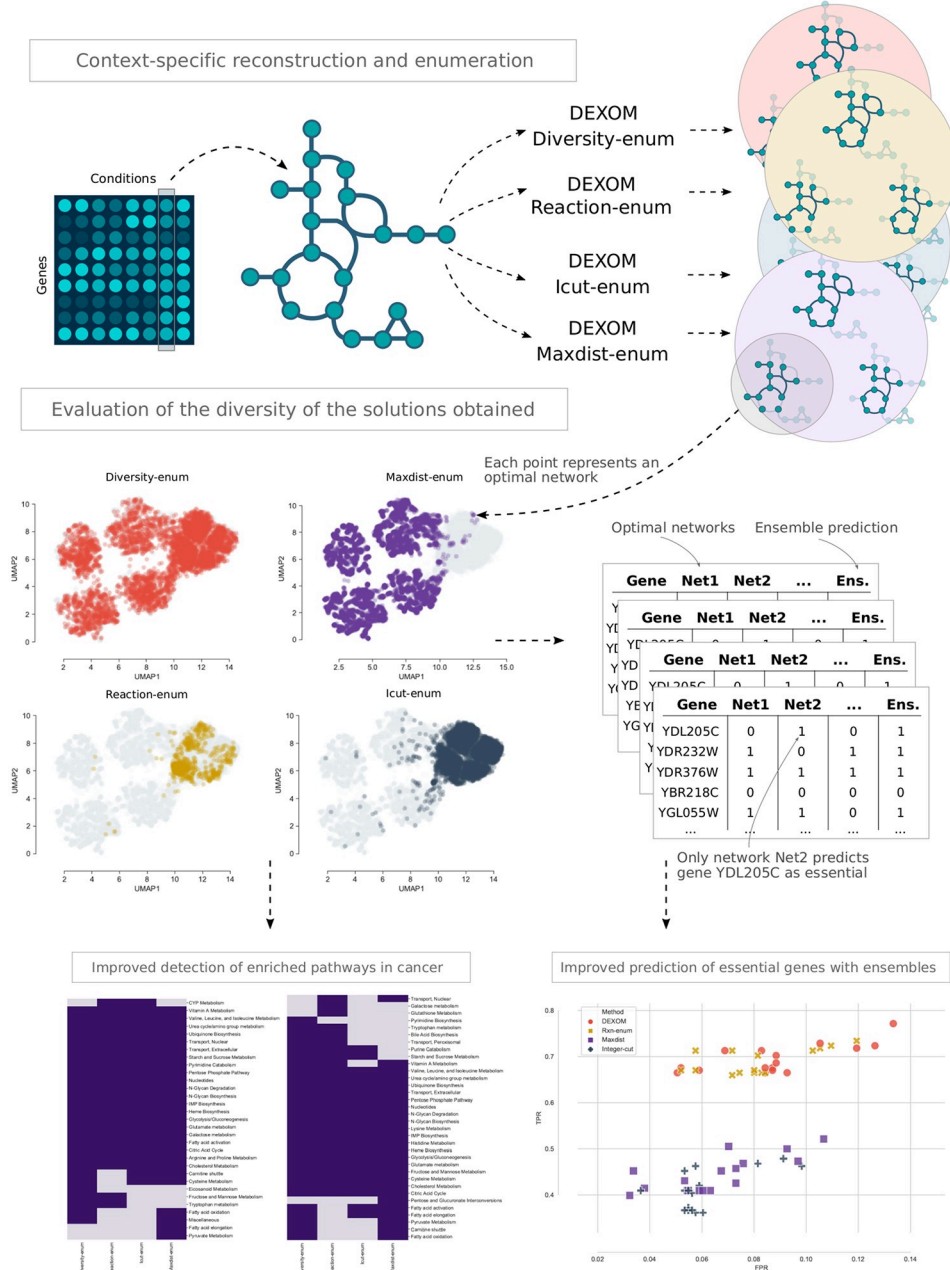

**Fig 1. Summary of the methods and validation.** Given an experiment (e.g., microarray data for different conditions), and a base genome-scale model, we use the four different methods included in `DEXOM` to enumerate the unknown set of multiple solutions (optimal context-specific metabolic networks, according to some objective function). Each method solves the same reconstruction problem but using a different strategy for discovering alternative solutions. Each set of optimal solutions is compared in terms of diversity, and projected into a 2D embedding to visualize which part of the space of optimal metabolic networks is explored by each method. In order to explore if a more diverse set of optimal solutions is biologically meaningful, we performed two evaluations with real data: 1) in-silico simulations of essential genes in yeast using ensembles of optimal networks; and 2) pathway enrichment in human cancer cells, using the whole set of discovered networks.

`Reaction-enum` is based on the idea of generating alternative solutions by single reaction changes. At each step, a different reaction in the model is picked and forced to be active or inactive to generate an alternative solution, which is kept only if the new solution is still optimal. The `Icut-enum` method is based on the idea of using integer-cuts as constraints to discard previously found solutions: at each step, a new solution is found and a new constraint is added to the original problem to discard this solution, making this solution not valid anymore. By progressively adding new constraints, new optimal solutions are found. The idea behind `Maxdist-enum` is to find at each step the most distant optimal solution with respect to the previous optimal solution, and using integer-cuts to avoid re-discovering the same distant solutions.

These three techniques, albeit simple and useful in many situations, have also limitations when it comes to discovering diverse sets of solutions. Based on the analysis of their limitations, we developed a fourth technique called `Diversity-enum`, a method that takes the best of the other three techniques without their disadvantages. Using experimental data for a particular condition and organism, `Diversity-enum` first construct an initial set of subnetworks by testing single variations of reactions that may or may not be present in the networks without affecting the optimality. This set is then incrementally expanded to find new optimal solutions by progressively maximizing the differences with other solutions previously found, increasing the distance at each step.

## Optimal context-specific metabolic network reconstruction

Here we consider the reconstruction of optimal context-specific metabolic networks as the selection of a subset of reactions from a global genome-scale metabolic network for a particular organism, in a way that maximizes the agreement with experimental data, i.e., reactions in the model with evidence of being active in a given context should be preserved, and reactions with evidence of being inactive should be removed from the model. The selection of this subset of reactions is also subject to flux-based constraints, which constrain the space of possible ways in which those reactions can be selected. More formally, given:

- $G = \{R, M, S\}$, an initial genome-scale metabolic network $G$ for a given model organism, where $R = \{R_1, \ldots, R_n\}$ is the set of reactions in the network, $M = \{M_1, \ldots, M_m\}$ is the set of metabolites, and $S$ is the stoichiometry matrix of size $m \times n$

- $f(x) : \{0, 1\}^n \to \mathbb{R}$, a linear objective function of the form $\mathbf{c}^T \mathbf{x}$ that returns a score for a candidate subset of reactions indexed by a binary vector $\mathbf{x} \in \{0, 1\}^n$, indicating whether reaction $R_i$ is selected or not, so that the subset of selected reactions from $R$ is defined as $R_c = \{R_i \in R | x_i = 1, \forall i \in 1 \ldots n\}$

The goal is to find the binary vector $\mathbf{x}$ (or equivalently the subset $R_c$) such that $f(\mathbf{x})$ is maximized. Reactions included in the $R_c$ set have to carry a non-zero flux under steady state conditions. This problem can be stated as a Mixed Integer Linear Programming (MILP) problem with the following form:

$$
\begin{aligned}
\max \quad & f(\mathbf{x}) = \mathbf{c}^T \mathbf{x} \\
\text{s.t.} \quad & \mathbf{S} \cdot \mathbf{v} = \mathbf{0} \\
& x_i * v_{\min,i} \leq v_i \leq x_i * v_{\max,i} \\
& \mathbf{v} \in \mathbb{R}^n, \quad \mathbf{x} \in \{0, 1\}^n
\end{aligned}
\tag{1}
$$

where $x_i \in \{x_1, \ldots, x_n\}$ are the binary variables representing if reaction $R_i$ is present or not, $v_i \in \{v_1, \ldots, v_n\}$ the variables representing the flux through each reaction $R_i$, and $v_{min}$ and $v_{max}$ the

lower and upper bounds for the flux through each reaction. Note that what is subject to optimization is the selection of the reactions but not the fluxes. Fluxes are constrained within some bounds $v_{min}$ and $v_{max}$, and forced to be in steady state ($S \cdot v = 0$). Reactions can be included ($x_i = 1$) only if they can carry some non-zero flux, and reactions not included ($x_i = 0$) are forced to carry a zero flux. In the following, we shall use this notation to introduce different MILP problems for context-specific reconstruction of metabolic networks.

The objective function $f(x)$ calculates the agreement between the experimental data and the selected reactions. One common strategy is to divide reactions in two disjoint sets based on experimental evidence, namely reactions associated with highly expressed enzymes ($R_H \subseteq R$) and reactions associated with lowly expressed enzymes ($R_L \subseteq R$), and then defining $f(x)$ as:

$$f(\boldsymbol{x}) = \sum_{i | R_i \in R_H} x_i + \sum_{i | R_i \in R_L} 1 - x_i \tag{2}$$

This is the strategy described in $\texttt{iMAT}$, in which the selection of one reaction in $R_H$ or the removal of one reaction in $R_L$ contribute in the same way to the score. Other strategies such as $\texttt{Fastcore}$, enforce the inclusion of all the reactions in $R_H$, and so $f(x)$ evaluates only the number of selected reactions in $R_L$ to minimize it.

In practice, the binary vector $\mathbf{x}$ is extended to account also for reversible reactions in the $R_H$ set that can be active carrying a negative flux. Also, a tunable parameter $\epsilon$ corresponding to the minimal amount of flux a reaction has to carry to be considered active is usually included in the optimization problem. In the original $\texttt{iMAT}$ formulation, a reaction $R_i \in R_L$ which is not selected (which carries no flux) has a value of $x_i = 1$ representing a match with the experimental data, and so Eq 2 simplifies to just $f(\mathbf{x}) = \sum_i x_i$. The full problem specification is described in Eq 3:

$$
\begin{aligned}
\max \quad & \sum_i x_i \\
\text{s.t.} \quad & S \cdot v && = && 0 \\
& v_i + x_i^+ (v_{\min,i} - \epsilon) && \geq && v_{\min,i} && \forall i \mid R_i \in R_H \\
& v_i + x_i^- (v_{\max,i} + \epsilon) && \leq && v_{\max,i} && \forall i \mid R_i \in R_H \\
& v_i + x_i^o \cdot v_{\min,i} && \geq && v_{\min,i} && \forall i \mid R_i \in R_L \\
& v_i + x_i^o \cdot v_{\max,i} && \leq && v_{\max,i} && \forall i \mid R_i \in R_L \\
\\
& v \in \mathbb{R}^n, \\
& x^+, x^-, x^o \in \{0,1\}^{|R|} \\
& x = (x^+, x^-, x^o) \in \{0,1\}^{3|R|}
\end{aligned}
\tag{3}
$$

It is important to remark that the methods presented here are general strategies for enumerating optimal metabolic network reconstructions, and therefore can be used with different base algorithms for the reconstruction, as long as they are implemented as MILPs. This means that the methods serve to enumerate $\texttt{iMAT}$-like solutions [7], $\texttt{Fastcore}$-like solutions [15], $\texttt{INIT}$-like solutions [14], or any other type of MILP-based reconstruction.

In the following sections, for practical reasons and without loss of generality, we use the original set of $\texttt{iMAT}$ constraints and objective function as the base MILP problem for network enumeration, since: 1) it relies on a MILP formulation, which can be easily adapted to optimally solve different optimization problems and objectives; and 2) the default objective

function optimizes a trade-off between the coverage of reactions associated with highly expressed genes and reactions associated with lowly expressed genes, which has been proven in practice a good general strategy that only requires gene expression data. This trade-off introduces flexibility in the optimization process, allowing us to predict that some reactions are not active even though they are associated with highly expressed genes, something important to account for post-transcriptional events.

## The problem of enumerating optimal metabolic networks

The enumeration problem arises naturally in context-specific reconstruction of metabolic networks due to the discrete nature of the selection of reactions and the imbalance between the available constraints and the complex topology of the networks, leading to an undetermined problem.

In order to better analyze the enumeration problem from a computational point of view, we use a Directed Acyclic Graph (DAG) network model. Directed Acyclic Graphs are commonly used for the analysis of biology networks in general [28]. By representing metabolic networks as DAGs, we can calculate in advance how many optimal solutions we can expect, and thus compare the techniques with a ground truth (i.e., the full set of optimal solutions that exist in a specific DAG) in an objective manner focusing specifically on the computational problem of enumeration. This is important because although the scope of application is biological, the technique is basically computational and requires a proper computational analysis of the problem under study. Fig 2 shows the generic DAG metabolic network with $L$ layers of $N$ metabolites. Each metabolite $m_{i,k}$ in layer $L_k$ is connected to each metabolite $m_{j,k+1}$ in $L_{k+1}$ by single reactions $R_{ijk} = (m_{i,k}, m_{j,k+1})$ with only one substrate and product. The model includes two extra metabolites $m_s$ as a source and $m_t$ as a sink node to centralize the import and export reactions and simplify the model. The number of total metabolites, including $m_s$ and $m_t$ is $2 + N \cdot L$, and the number of total reactions is $2N + N^2 \cdot (L - 1)$.

In this example, we want to extract the context-specific metabolic network, given the following conditions:

• $\sum_i |v_i| > 0$, i.e., there is a non-zero steady state flux from $m_s$ to $m_t$. This is commonly assumed in order to avoid having an empty network.

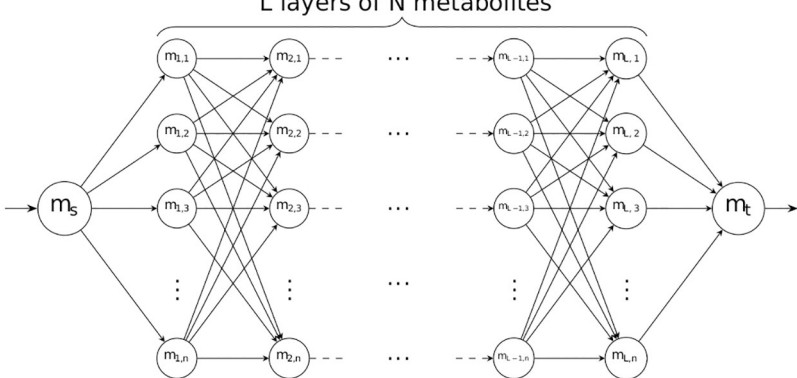

**Fig 2. Directed Acyclic Graph (DAG) metabolic network model.** This figure illustrates the DAG metabolic model that we use to analyze the computational issues related to the enumeration of optimal context-specific network reconstructions using MILP-based reconstruction methods. The metabolic network is divided into $L$ layers, each layer containing $n$ metabolites.

- $R_H = \emptyset$, $R_L = R$, i.e., there are no reactions associated to highly expressed enzymes, and all the reactions are associated with lowly expressed enzymes.

It can be seen that a metabolic network with optimal $f(\mathbf{x})$ in this case is the one that carries flux from $m_s$ to $m_t$ using the minimum number of reactions (since they are all in the $R_L$ set), which corresponds to a shortest path from $m_s$ to $m_t$. Since there are no loops in the network, the shortest length for the path is $L + 2$ (including the path from $m_s$ to $L_1$ and from $L_N$ to $m_t$). This also implies that there is no single solution, but instead any path from $m_s$ to $m_t$ is an optimal solution, i.e., a context-specific reconstruction network with optimal $f(\mathbf{x})$ given the previously defined conditions. Since there are $N$ different paths to go from any metabolite in layer $L_j$ to any metabolite in layer $L_{j+1}$, that makes $N^L$ possible optimal networks in this particular example, that is, the number of possible optimal solutions in this example grows exponentially with the number of layers. Note also that since the number of reactions for a fixed number of metabolites grows linearly with the number of layers, the number of possible solutions grows also exponentially with the number of reactions.

This example illustrates that there are instances of the enumeration problem for which the number of optimal solutions grows exponentially with the size of the network. Thus, in general, enumerating the full set of optimal metabolic networks can be impractical, especially considering the size of networks such as Recon 3D [29] with 13,543 reactions, or the recent Human1 network [30] with around 13,000 reactions.

More formally, it can be shown that the enumeration of all optimal metabolic networks is a type of *vertex enumeration problem*. Let $M_P$ be the general MILP problem for context specific reconstruction using a GSMN with $n$ reactions and with objective function $\mathbf{c}^\mathbf{T}\mathbf{x}$ that we want to maximize, as defined in Eq 1. Let $\Omega$ be the set of all 0/1-vectors representing the feasible solutions for the MILP $M_P$ that satisfy all the constraints defined in Eq 1. From a geometric point of view, the space of possible networks can be viewed as vertices of the hypercube $C_n = \{0, 1\}^n$, and the set of feasible solutions $\Omega$ as a subset of vertices of $C_n$, where its convex hull is a 0/1-polytope $P$, that is, $P = conv(\Omega)$. Let $z^*$ be the optimal value of $M_P$, i.e., $\forall \mathbf{x} \in \Omega, \mathbf{c}^\mathbf{T}\mathbf{x} \leq z^*$. We are interested in the set of all optimal feasible solutions $\Omega^* := \{\mathbf{x}^* \mid \mathbf{x}^* \in \Omega \wedge \mathbf{c}^\mathbf{T}\mathbf{x}^* = z^*\}$, where $P^* = conv(\Omega^*) \subseteq P$ is the 0/1-subpolytope of interest in $\mathcal{H}$-representation (as the intersection of half spaces defined by all the constraints) from which we want to obtain the $\mathcal{V}$-representation, that is, the set of vertices as vectors of 0/1 coordinates (the optimal context-specific networks), which is the definition of the *vertex enumeration problem*.

Vertex enumeration [31] is a classical problem in the field of combinatorial optimization for which some specific techniques were proposed [32]. For the special case of 0/1-polytopes [33], some notable approaches are Binary Decision Diagrams [34–38], tree search-based methods [39, 40] and techniques based on branch-and-bound and cutting planes, extensively exploited in academic/commercial solvers such as IBM CPLEX and Gurobi. In fact, as a general enumeration mechanism, these solvers incorporate the concept of a pool of optimal solutions, in which the tree of feasible solutions continues to be explored until a specific number of optimal feasible solutions have been found.

However, as discussed before, the number of optimal metabolic networks for a given problem can be extremely large, and so classical vertex enumeration techniques are not suitable for this task. One reason is that, given the potential vast number of possible solutions and a fixed amount of time to generate a variety of optimal solutions, there is no guarantee that these methods will generate a diverse set of solutions. In fact, the opposite is more likely: similar solutions (e.g., small variations in reactions on the same pathway) will probably be closer in the search space. Also, due to symmetries introduced by loops and other patterns in metabolic networks, chances are that the enumeration gets trapped performing enumeration in small

dense regions of the search space that can be more related to artifacts than to solutions with true biological meaning.

In the following sections we present four enumeration strategies and analyze their advantages and drawbacks. It should be noted that we limited to a set of generic techniques that can be implemented on top of general MILP solvers and can be easily integrated in the existent pipelines for network reconstruction. One disadvantage of this is that each solution is obtained by constructing and solving a new MILP problem. Ad-hoc search strategies for the enumeration of MILP solutions based on custom branch-and-cut methods or more advanced tree search exploration, although they might be more efficient in some situations, are out of the scope of this work.

## Enumeration of optimal networks by inclusion or exclusion of reactions (`DEXOM Reaction-enum`)

A simple way to generate alternative optimal metabolic networks can be achieved by directly manipulating the flux bounds of each reaction to force it to carry some positive flux, some negative flux (if reversible), or no flux, as in [20, 21]. The original method traverses all the reactions in the model testing forward (or backward flux if the reaction is reversible) or blocking flux in order to generate a new solution with a different activation for each reaction. Solutions that are still optimal after the modification are added to the set of optimal solutions. This method has however two major limitations: 1) it only explores variations in single reactions (if they can be active or inactive in an optimal solution), leaving the vast space of combinations between reactions completely unexplored; and 2) it generates many duplicated solutions, wasting computation time.

We included this basic idea in `DEXOM` as a simple mechanism to generate alternative solutions, under the name of `Reaction-enum`, with some modifications and further options that can be used to reduce some of the limitations in its basic form. One option that can be enabled to alleviate the problem of generating duplicated solutions consists in tracking the activation or inactivation of each reaction in the set of alternative optimal networks during the search process. If forcing the flux through a reaction $R_{ijk}$ results in an optimal sub-network with another reaction $R_{i,j+1,k+1}$ active, then there is no need to force flux through $R_{i,j+1,k+1}$ as it is not guaranteed that this operation is going to generate a new sub-network (unless the solver is adjusted to increase randomness in the solutions returned).

One advantage of the `Reaction-enum` method is that it tests every reaction in the model to see if its presence or absence affects the quality of the solution. This generates alternative networks with modifications in every possible pathway of the metabolic network, which makes this technique a good starting point for more advanced enumeration methods (for example, to generate a set of initial candidate optimal solutions).

## Exhaustive enumeration of optimal networks (`DEXOM Icut-enum`)

One simple way to perform a full enumeration of the set of optimal networks is by adding integer-cuts, linear constraints that can be added to the original problem to remove already visited solutions from the set of feasible solutions. This method, which has been already used to enumerate solutions in general for MILP problems [41], can be used as well as a mechanism to enumerate alternative metabolic networks. We adapted this technique for enumeration of context-specific reconstructions under the name of DEXOM Icut-enum. Starting with a default optimal solution $\mathbf{x}^*$ to the MILP problem defined in Eq 3, a new solution is generated by adding a new constraint to the original problem to cut $\mathbf{x}^*$ from the set of feasible optimal solutions. This process is repeated for each new solution, adding a new constraint per solution. A new

solution is accepted if there is at least one different reaction in the candidate sub-network, that is:

$$\sum_i |x_i - x_i^*| \geq 1 \tag{4}$$

Although this constraint is not linear due to the absolute value, it can be linearized by considering separately the ones from the zeros. Two solutions are equal if they have the same set of active reactions and the same set of inactive reactions. Thus, for each $x_i^* = 1$, we expect to have $x_i = 1$, and for each $x_i^* = 0$, we expect $x_i = 0$ if both the previous solution and the candidate are equal. Under this situation, summing up all the ones from $x_i$ for which $x_i^* = 1$ should be equal to $\sum(x_i^*)$ (except if there is one or more differences), and in the same way, summing up all the zeros from $x_i$ for which $x_i^* = 0$ should be equal to zero. If this does not happen, then there is some difference between the candidate solution $x_i$ and a previous optimal solution $x^*$. More formally, the linearization of Eq 4 can be written as:

$$\sum_{i \in A} x_i - \sum_{i \in B} x_i \quad \leq \quad \left( \sum_i x_i^* \right) - 1$$
$$A = \{i \,|\, x_i^* = 1\}, B = \{i \,|\, x_i^* = 0\} \tag{5}$$

By adding this constraint for each new $\mathbf{x}^*$ returned by the solver, we exclude all the previous solutions that have been found so far. The generation of new solutions stops when the problem becomes infeasible, that is, there are no more feasible optimal solutions. Note that this cut can be modified to cut feasible optimal solutions that differ in more than one reaction, i.e., to cut solutions that are at some specific Hamming distance.

The advantage of this method is that it enumerates all possible solutions since it removes one by one every optimal feasible solution. It is straightforward to see that this method enumerates all the feasible optimal solutions by observing that: 1) each cut removes one optimal solution; 2) the number of constraints that are added grows monotonically with every new optimal solution; and 3) the number of solutions is finite. Let us assume that for a given problem, the set of optimal feasible solutions $\Omega^*$ contains $N$ different solutions, i.e., for every pair $\{x^*, y^*\} \subset \Omega^*, \sum_i |x_i^* - y_i^*| \geq 1$ (there is at least one different reaction between any two optimal solutions). For the sake of the proof, let us assume that after $N$ steps of the algorithm, and after adding $N$ integer cuts, one per optimal solution, the last MILP problem is still feasible, i.e., solving it returns a solution $z^*$, thus: 1) $z^*$ is different to any other solutions in at least one reaction, which means that there are at least $N + 1$ solutions, contradicting the initial assumption; or 2) $z^*$ is a duplicated solution, that is, there exist a solution $x^* \in \Omega^*$ such that $\sum_i |z_i^* - x_i^*| = 0$, which contradicts the definition of the integer cut.

However, in practice, it is not possible to enumerate the entire space of solutions due to the potential number of possible optimal solutions. Although this technique can be also used to generate a sample of optimal solutions (stopping the search after a desired number of solutions was found), the method is not well suited for this task since: 1) the number of constraints grows linearly with the number of solutions, which increases the computational difficulty with each new solution; 2) the algorithm can get trapped enumerating solutions in a small region of the whole space of possible optimal solutions, and so diversity in the set of solutions is not guaranteed; 3) even if a new optimal solution exists, due to numerical instabilities or precision errors, the search process can prematurely stop at the first incorrectly detected infeasible problem.

## Enumeration of optimal networks with maximum dissimilarity (`DEXOM Maxdist-enum`)

Another strategy for the enumeration of optimal solutions is to search the most dissimilar metabolic network to a previous optimal one. This idea, already explored in the context of Integer Programming problems [25, 42], has been also proposed for metabolic networks [22]. The strategy requires to solve a bilevel optimization problem in which the inner optimization problem solves the original problem and the outer optimization maximizes dissimilarity. This particular bilevel optimization can be implemented as a standard MILP problem, by introducing a constraint that corresponds to the original objective function. First, an optimal solution $\mathbf{x}^*$ with optimal score $f(\mathbf{x}^*) = z^*$ is calculated using the problem defined in Eq 3, and then the original objective function is replaced by the minimization of a function $g(\mathbf{x}, \mathbf{x}^*)$ which measures the similarity between the candidate solution $\mathbf{x}$ and a reference optimal solution $\mathbf{x}^*$. In order to guarantee that the new solution to this new problem is still optimal in the original problem, a new constraint $f(\mathbf{x}) = \sum_i(x_i) = z^*$ has to be added to preserve optimality.

Although the idea of returning the most dissimilar optimal network is interesting, one of the limitations is that it can easily oscillate between a small set of optimal networks that are the most distant to each other, since only the previous optimal solution is discarded. Consequently, the technique also does not guarantee to obtain all the possible solutions. We have introduced a modification to the original idea to break this pattern and allow the complete enumeration of solutions by adding integer cuts. This modification prevents trivial oscillations between already visited solutions and enumerates the space of solutions starting from the most extreme differences. We refer to this technique as `Maxdist-enum`.

The objective function $g$ of the `Maxdist-enum` method can be defined as the minimization of the overlapping of ones between $\mathbf{x}$ and $\mathbf{x}^*$. Note that the optimality constraint guarantees that the solutions must have the same number of ones (same score), and so removing one overlap (for example by not including a reaction in $R_H$ which is present in the reference solution) has to be compensated by including another reaction in the set of $R_H$ not present in the reference solution, or by removing one reaction in the $R_L$ set which is present in the reference solution, in order to preserve the original optimal score. Minimization of the overlapping of ones between $\mathbf{x}$ and $\mathbf{x}^*$ with this constraint is essentially the same as finding the most extreme vertices of the 0/1-polytope of feasible optimal solutions using the Hamming distance.

$$
\begin{aligned}
\min_{\mathbf{x}} \quad & g(\mathbf{x}, \mathbf{x}^*) = \sum_{i|x_i^*=1} x_i \\
\text{s.t.} \quad & \mathbf{S} \cdot \mathbf{v} = 0 \\
& \sum_{i \in A} x_i - \sum_{i \in B} x_i \leq \left(\sum_i x_i^*\right) - 1 \\
& \sum_i x_i = z^* \\
& v_i + x_i^+(v_{\min,i} - \epsilon) \geq v_{\min,i} & \forall i \mid R_i \in R_H \\
& v_i + x_i^-(v_{\max,i} + \epsilon) \leq v_{\max,i} & \forall i \mid R_i \in R_H \\
& v_i + x_i^o \cdot v_{\min,i} \geq v_{\min,i} & \forall i \mid R_i \in R_L \\
& v_i + x_i^o \cdot v_{\max,i} \leq v_{\max,i} & \forall i \mid R_i \in R_L
\end{aligned}
\tag{6}
$$

The expected behavior of this algorithm is the following: starting from the default solution $\mathbf{x}^*$, the search process generates the most distant network with the same optimal score. This process is repeated, changing the $\mathbf{x}^*$ in each iteration to the one previously found, pushing

away the search to the boundaries of the space until the most distant networks in the space of optimal solutions are discovered. The `integer-cut` constraint prevents search loops, and so once the extremes are found, the distance of new solutions decreases progressively.

This method has also limitations that may prevent its use for generating a diverse sample of optimal solutions. Concretely, even though the integer-cut constraint prevents generating repeated solutions, the density of similar metabolic networks at the boundaries can be large enough to never explore other areas. This increases the risk of ending up oscillating between a small group of clusters of networks with a large inter-cluster distance but a very small intra-cluster distance. In addition to this, the method is computationally more expensive than the previous ones.

## Diversity based extraction of optimal metabolic networks (`DEXOM Diversity-enum`)

Based on the previously identified problems and improvements for each method, we also propose a novel algorithm to generate a set of diverse optimal metabolic networks, combining the advantages of the techniques described before. The basic steps of `Diversity-enum` are:

1. Generate an initial set of optimal solutions using the `Reaction-enum` method with integer cuts to avoid duplicated solutions.

2. Pick an initial solution $\mathbf{x^{(0)}}$ from this set.

3. Find a new alternative solution maximizing the differences with respect to some $n$ random reactions that are present in the initial solution (this is, find the maximum dissimilarity optimal network with respect to only those $n$ reactions). The number of reactions that are selected ($n$) increases over time, starting with only 1 reaction (alternative solutions should be different in at least one reaction), until $n$ = number of present reactions in $\mathbf{x^{(0)}}$ (i.e., maximize differences with respect to all the selected reactions in the initial solution, which is essentially the same as in the `maxdist-enum` method). In this way, `Diversity-enum` behaves at the beginning more like `Reaction-enum`, increasing progressively the distance, until it behaves like `Maxdist-enum` method. The speed of this transition is controlled by the parameter $d_s$ (see Alg. 1).

4. Set the new solution $\mathbf{x^{(1)}}$ as the new initial solution and repeat from 3 until the desired number of solutions has been reached or until there are no more solutions.

**Algorithm 1** `Diversity-enum` algorithm
```
 1: procedure DIVERSITY_ENUM(iters, d_s, f)
 2:   x_r^(0),...,x_r^(k) ← initial solutions with the reaction − enum method
 3:   i ← 0
 4:   x^(i) ← x_r^(k)
 5:   z* = f(x^(i))
 6:   while i < iters and f(x^(i)) = z* do
 7:     y^(i) ← vector of 0s of same size as x^(i)
 8:     pick_prob ← 1 − exp(d_s, i) # where exp(a, b) = a^b
 9:     for j | x_j^(i) = 1 do
10:       sample u ∼ U(0,1)
11:       if u ≤ pick_prob then
12:         y_j^(i) ← 1
13:     s ← solve maxdist MILP min_s g(s, y^(i)) (Eq 6)
14:     i ← i + 1
15:     x^(i) ← s
16: return x_r^(0),...,x_r^(n),x^(1),...,x^(i).
```

A detailed version of the algorithm is described in Alg. 1. `Diversity-enum` combines the advantages of the previous techniques. It starts computing an initial set of solutions using the `Reaction-enum` method avoiding duplicated solutions. This guarantees that single variations of reactions across all pathways are explored, as long as this initial set of solutions is large enough (i.e., all reactions of the network are traversed to generate alternative solutions). Then, starting for any solution from this initial set, the algorithm explores solutions in the vicinity of the selected solution, using it as a *template*, for which only a subset of the reactions present in the selected solution are used to maximize the distance to the new solution. The more reactions that are selected to maximize the distance, the more different the new solution found will be from the selected one. The number of selected reactions from the template at each iteration (i.e., the distance of the next solution with respect the previous one) is controlled by the parameter $d_s \in [0, 1]$. For example, using a $d_s$ value close to one (e.g. $d_s = 0.99$), the distance of the solution obtained at iteration 70 with respect the previous one (obtained in the iteration 69), is going to be $1 - 0.99^{70} \approx 0.5$, that is, the expected distance of the next optimal solution with respect the previous one is half of the possible maximal distance. At iteration 1,000, this value is $1 - 0.99^{1000} \approx 1$, and so the algorithm is now searching for the most distant solution, as `Maxdist` does. Using a higher value of $d_s$ (e.g. $d_s = 0.999$) makes this transition from near to far solutions slower, since this time the value at iteration 70 is only $1 - 0.999^{70} \approx 0.07$. It should be noted that if $d_s = 0$, after computing the initial set of solutions, `Diversity-enum` behaves exactly as `Maxdist-enum`. By default, the value of this parameter has been set to $d_s = 0.995$, and all experiments performed, unless otherwise indicated, have been done with this value.

Some preliminary experiments that we carried out suggest that it is preferable to start with the initial set of solutions using the `Reaction-enum` method and expand it by progressively looking for more distant solutions, rather than the other way around. The reason is that if we start with the most extreme solutions, as we progressively decrease the distance, the effect we get is to explore solutions that are closer to each other but still located in the extremes of the space, and still far from the initial solutions.

## Measuring diversity

Given the unknown volume of the 0/1-polytope comprising the optimal solutions, it is not possible to directly estimate its size without sampling solutions from it. In order to measure how diverse are the set of solutions obtained with different methods, we need to rely instead on indirect measures. Since solutions are indexed by 0/1 coordinates, one reasonable metric to use is the Hamming distance:

$$\delta_h(\boldsymbol{x}, \boldsymbol{y}) = \frac{1}{|\boldsymbol{x}|} \sum_{i=1}^{n} |x_i - y_i| \tag{7}$$

For each pair of solution vectors $\boldsymbol{x}, \boldsymbol{y} \in \{0, 1\}^n$ obtained from the set of optimal solutions $\Omega^*$, we compute the Hamming distance (i.e., how many reactions are different between any two solutions) and we average across all the distances between any two solutions to obtain the *average pairwise distance* $\bar{\delta}_h$. One way to promote diversity is to maximize this measurement: between two different sets of optimal solutions (of a similar size), the set with a larger average pairwise distance contains solutions that are, on average, more diverse. However, relying only on the average pairwise distance might not be informative enough in some situations, since two groups of solutions that are very different between groups but very similar within groups, can have a large average pairwise distance driven by the distance between groups, even thought

the diversity is low within groups. Under these circumstances, it is easy to have the false impression that the set of solutions is diverse, but instead it will contain only the two initial different solutions with very small variations.

To discriminate better between these situations, we use also the *average nearest neighbor distance* $\bar{\delta}_h^{nn}$ defined as:

$$\bar{\delta}_h^{nn}(O) = \frac{1}{|S|} \sum_{x \in O} \min_{y \in O \setminus \{x\}} \delta_h(\boldsymbol{x}, \boldsymbol{y}) \tag{8}$$

That is, for each optimal solution in the solution set $O \subseteq \Omega^*$ obtained with some enumeration method, we measure the distance to all other solutions and we take the distance to its closest solution (nearest neighbor). Then, we average all those distances to have the average nearest neighbor distance.

The average nearest neighbor distance measures how *spread* the solutions are. We want solutions that are spread to cover a wider range of different solutions and avoid the enumeration of clusters of very similar solutions.

Considering these two metrics, we can devise four situations when comparing the solution sets obtained by different methods:

- **Lower** $\bar{\delta}_h$ and **lower** $\bar{\delta}_h^{nn}$: this situation corresponds to a low diversity. Solutions are close together and sampled from a small region of the search space.

- **Larger** $\bar{\delta}_h$ and **lower** $\bar{\delta}_h^{nn}$: low dispersion of the solutions, even though solutions are distant from each other.

- **Lower** $\bar{\delta}_h$ and **larger** $\bar{\delta}_h^{nn}$: solutions are dispersed but only in a smaller region of the search space.

- **Larger** $\bar{\delta}_h$ and **larger** $\bar{\delta}_h^{nn}$: better diverse set of solutions in which solutions are scattered across the space of optimal networks.

Although simple, these metrics provide an idea of how different the solutions enumerated by the methods are.

## Essential gene prediction and metabolic network ensembles

Context-specific metabolic networks can be used to make predictions about the metabolism of a cell or tissue in a specific experimental condition. Of a particular interest is the prediction of essential genes. An essential gene is a gene whose totally or partially inactivation prevents the organism to growth or survive. Some genes are absolutely required for survival, whereas other genes are conditionally essential, meaning that they are essential depending on the environmental conditions. For example, the gene ARG2, which encodes glutamate N-acetyltransferase —a mitochondrial enzyme that catalyzes the first step in the biosynthesis of the arginine— is annotated as a essential gene in *Saccharomyces cerevisiae* (https://www.yeastgenome.org/locus/S000003607) only in the absence of arginine in the medium.

Many essential genes that are related to metabolism (those related to enzymes) can be predicted using metabolic networks. However, conditionally essential genes are particularly hard to predict since they cannot be predicted without integrating experimental data or knowledge related to the condition. Context-specific metabolic networks are able to predict them indirectly, by extracting first the sub-network which is most consistent with the experimental data. After removing all the reactions that are predicted to be inactive in a given context,

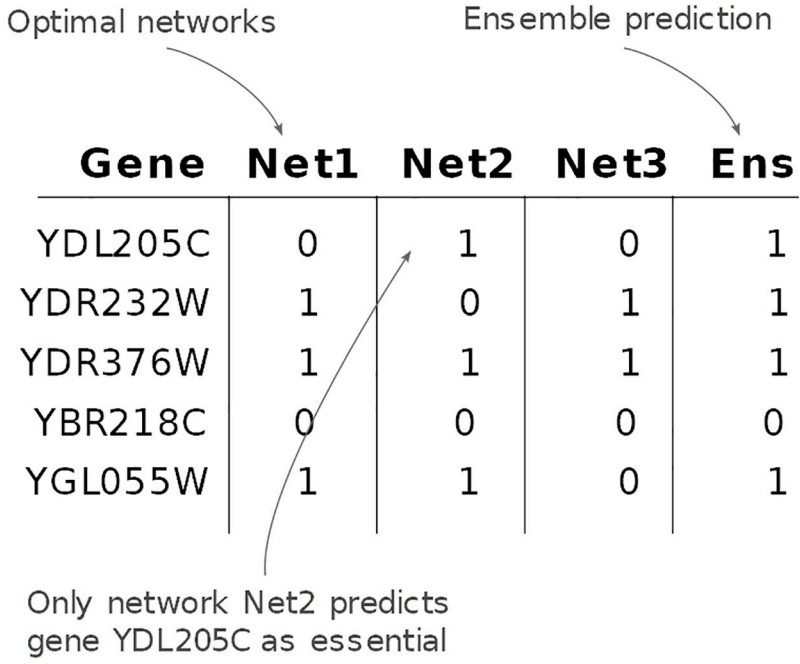

**Fig 3. Example of a metabolic network ensemble.** Predictions of the essential genes produced by Net1, Net2 and Net3 are combined by performing a logical OR.

conditionally essential genes that were not essential in the generic network might be now essential in the reconstructed network.

Predictions of essential genes using metabolic networks can be done by comparing the maximum flux through the biomass reaction —an artificial reaction that encodes the minimum requirements of the organism to sustain a basic metabolic activity— using Flux Balance Analysis (FBA) [43] before and after knocking out a gene in the metabolic network. If the flux through the biomass reactions is below a certain threshold after KO (e.g., below 1% with respect to the wild-type) then the gene is considered essential.

However, as explained before in this section, it is common to find more than one optimal context-specific metabolic network for a given condition, each one representing a different hypothesis of the metabolic state. Each network may predict different essential genes. Since all networks fit the experimental data equally well, there is no clear way to decide a priori which of these predictions may be true. In this situation, a reasonable strategy is to consider that if a network predicts a gene to be essential, then the ensemble decides that the gene is essential, in order to maximize the number of true essential genes (at expenses of increasing the false positives), similar to what has been done in [24] with Gap-Filling methods.

Fig 3 shows an example of how the procedure works. For each gene, a KO is simulated by maximizing the flux through the biomass reaction after knocking out the reaction or reactions associated to the gene (based on the Gene-Protein-Reaction rules), using the `singleGeneDeletion` method from the COBRA Toolbox [27]. If the ratio between the KO and the wild type is below 0.01 (flux after KO below 1%), the gene is classified as essential. This process is repeated for all genes and for all optimal networks, and then results are combined by performing a logical OR of the predictions across networks.

After obtaining the predictions for each gene, the True Positive Rate (TPR, sensitivity) and the False Positive Rate (FPR, 1-specificity) are calculated by comparing the predictions against

the true essential genes for *Saccharomyces cerevisiae* (included in the repository of the code), and applying the following formula:

$$TPR = \frac{TP}{TP + FN} \quad (TP = \text{True Positives}, FN = \text{False Negatives}) \tag{9}$$

$$FPR = \frac{FP}{FP + TN} \quad (FP = \text{False Positives}, TN = \text{True Negatives}) \tag{10}$$

## Data pre-processing

One common step prior to any metabolic reconstruction is pre-processing the experimental data to map it onto the metabolic networks. The way in which this data is pre-processed also depends on the objective of the reconstruction and the type of data used (generally gene expression data). A common approach is to use gene expression data to classify reactions into two groups: reactions for which there is experimental evidence of being active for a given condition, and reactions for which there is not enough evidence.

A simple method that is frequently used for this purpose is based on a prior classification of genes using quantile thresholds on the normalized gene expression [19]. In this way, genes whose expression levels are above or below certain quantiles are classified as highly or lowly expressed genes. For example, a thresholds specified as [0.25, 0.75] means that genes whose value are below the 25th percentile are classified as lowly expressed, and genes above the 75th percentile are classified as highly expressed. Afterwards, genes are mapped onto the metabolic network using the Gene-Protein-Reaction rules defined in the GSMN in order to get the reactions associated with highly expressed or lowly expressed genes.

Fig 4 shows an example for a threshold of [0.10, 0.90] on the normalized microarray gene expression of the *Saccharomyces cerevisiae* under aerobic conditions [18, 26]. In this example, genes whose expression levels fall above the upper threshold (around a normalized gene

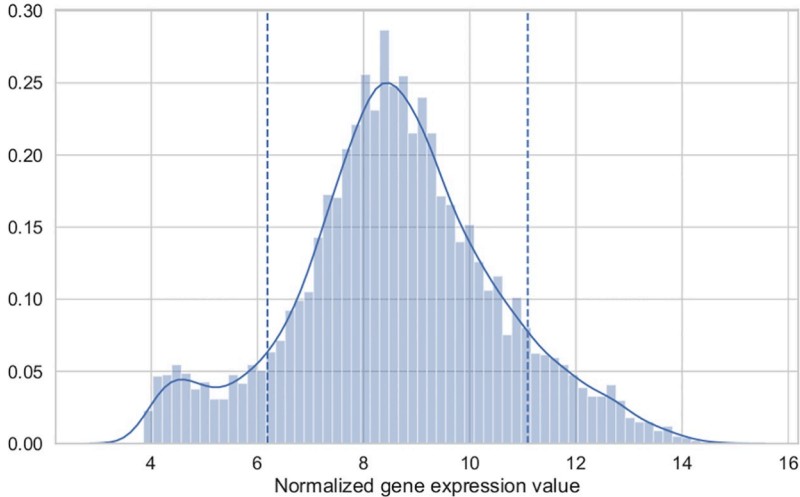

**Fig 4. Example of gene expression thresholds.** The example shows the quantile thresholds [0.10, 0.90] (indicated with dashed lines) on the normalized gene expression levels (RNA-seq) from *Saccharomyces cerevisiae* under aerobic conditions (20.9% oxygen levels) from [18, 26]. Genes above the upper threshold are classified as highly expressed genes, whereas genes below the lower threshold are classified as lowly expressed genes.

expression of 11) are considered highly expressed, whereas genes below a normalized gene expression around 6 are classified as lowly expressed genes.

As mentioned before in the section *Methods*, for practical reasons in this work we use the iMAT reconstruction objective as the base reconstruction problem for enumeration. This means that the results we enumerate are those that achieve the optimal trade-off between selection of reactions associated with highly expressed genes and removal of reactions associated with lowly expressed genes.

We use this threshold-based method for the classification of reactions based on gene expression levels because of its simplicity and widespreadness, but other methods could be used instead, for example StanDep [44] or Barcode [45] (for Affymetrix microarray data). Changing the method changes the set of optimal solutions to the problem, but does not eliminate the problem associated with the enumeration. Analyzing the correctness of the pre-processing technique is beyond the scope of this work, since the problem of enumeration is independent of the pre-processing method (multiple optimal solutions can still exist regardless of the method used).

## Results

In Section *Methods* we show how the problem of context-specific metabolic network reconstruction is subject to significant variability due to the vast number of possible optimal metabolic networks that explain the same experimental data. This variability makes the interpretation of the metabolism using a single metabolic network not very reliable, since many equally valid alternative hypotheses are disregarded.

In this section, we analyze the performance of each of the methods implemented in DEXOM to generate a diverse sample of optimal metabolic networks, assuming that in practice it is not possible to fully enumerate the total unknown space of optimal solutions, as is generally the case. The evaluation is divided into three parts.

First, we evaluate the diversity of the set of solutions discovered with each method in two scenarios: 1) when the true number of possible solutions is known (known ground truth), using the simple Direct Acyclic Graph model introduced in Section *Methods*; and 2) when the number of possible solutions is not known, using the Yeast 6 GSMN [46] as a biological realistic metabolic network. For the second case, we select random sets of highly expressed and lowly expressed genes from the Yeast 6 model to generate problems in which the total number of optimal solutions is not known a priori, and we compare the samples generated with each method in terms of diversity.

Second, we evaluate the predictive capabilities of each method for in-silico prediction of essential genes. Using real gene expression data for *Saccharomyces cerevisiae* under aerobic conditions [18, 26] and the Yeast 6 model [46], we enumerate thousands of optimal networks with each method and we asses the performance by predicting which genes are essential using both the individual networks and ensembles of networks constructed by combining the predictions of the individual networks.

Finally, we use gene expression data from four different human cancer cell lines and we reconstruct many optimal networks per cell line using different combinations of gene thresholds and methods. We compare the ability of each technique to discover alternative hypothesis of the metabolic state of the cells by performing pathway enrichment on the set of optimal solutions.

### Diversity-enum explores a wider region of the optimal network space

We measure how well each method performs to generate diverse samples of optimal solutions. To do so, we generate samples of fixed size with each method and we measure the diversity of

the sample using the average Hamming distance and the average nearest neighbor that were introduced in Section *Methods*. We consider two different scenarios: 1) obtaining a sample of optimal metabolic networks in a simulated scenario where the number of total optimal solutions is known; and 2) obtaining a sample of optimal solutions in realistic scenarios where the total number of optimal solutions is unknown.

**Evaluation in a simulated scenario with a known number of possible optimal solutions.** One of the difficulties of measuring the diversity of the solutions obtained by different methods is the absence of a ground truth to compare with, as the full set of optimal solutions is in general not known. However, the DAG network model introduced before can be used as a simple ground truth generator, since the full set of optimal solutions is easy to determine.

In order to assess the coverage and diversity of a sample of optimal networks, we used the DAG network model with 5 layers and 4 metabolites per layer (74 reactions and 22 metabolites in total), which contains a total of 1,024 optimal metabolic networks. The different methods were used to sample from 1 to 250 optimal solutions (around 1/4 of the total set of possible optimal solutions).

Fig 5 shows a low-dimensional projection of the 250 optimal solutions obtained by each method, where each point is an optimal metabolic network encoded as a binary vector. The grey points correspond to the total of 1,024 optimal solutions that exist for this example.

The `Reaction-enum` method shows a low coverage of the space of optimal solutions, enumerating only a 7% of the full space of optimal networks. This is due to the fact that the `Reaction-enum` method changes the bounds of each reaction in the network independently from each other. Since each reaction participates in many optimal solutions, the `Reaction-enum` can obtain only a subset of all possible optimal networks, missing a large fraction of optimal metabolic networks that cannot be recovered with this method.

Qualitatively speaking, the 250 solutions obtained with the `icut-enum` method are not as spread as the ones obtained with `Diversity-enum` and the `Maxdist-enum` method. Differences between `Diversity-enum` and `Maxdist-enum` are less obvious and hard to appreciate in a low dimensional embedding in this example.

In order to have a better picture of the diversity of the solutions, we calculated the evolution of the distances $\bar{\delta}_h$ and $\bar{\delta}_h^{nn}$ for each method. We repeated the process 30 times to obtain different samples of 250 solutions. The results for the 30 independent runs are shown in Fig 5E and 5F. The average over the 30 runs is represented with a dashed line.

These figures show in a more clear way how `Diversity-enum` obtains the most diverse set with respect the other methods after 150 optimal solutions were enumerated, surpassing the `Maxdist-enum` method. It can be seen how the behavior of the algorithm in terms of diversity changes dramatically after the initial solution set is calculated, around solution 50 (this effect is controlled by the *ds* parameter described in the `Methods` section). At this point, `Diversity-enum` starts to increase the distance progressively, looking for more and more distant solutions, which is reflected in the increase of both $\bar{\delta}_h$ and $\bar{\delta}_h^{nn}$. In contrast, `Reaction-enum` obtains sets of solutions with a very poor diversity. After calculating 74 solutions, the method cannot generate new optimal networks (since there are only 74 non reversible reactions in the network), and the solution set stops growing. Since the `Reaction-enum` generates solutions by modifying the constraints of each reaction, one at a time and independently of each other, solutions are mostly concentrated in a concrete region of the space of possible solutions, which corresponds to solutions that are similar to each other. The `Maxdist-enum` method shows at the beginning of the search the largest distance, since the solutions are generated by finding extreme differences. After an initial set of 25 optimal solutions, the average distance stops increasing, but the average nearest neighbor distance continues to decrease.

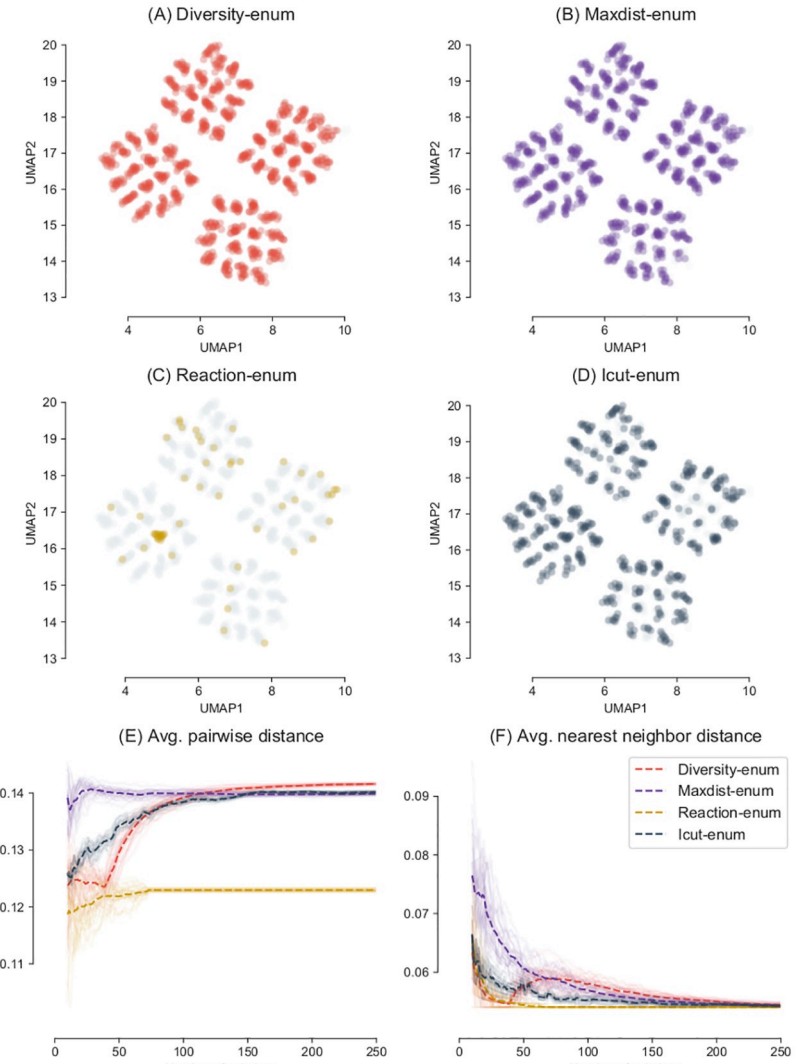

**Fig 5. Space of optimal solutions of the DAG network problem explored by each method.** Low dimensional representation of the optimal networks enumerated with different methods. Each method was used to explore a maximum of 250 optimal solutions, out of the 1,024 existent solutions (grey points). Each point represents an optimal metabolic network as a binary vector projected in 2D using UMAP with Hamming distance and 30 neighbors. Both `Diversity-enum` and `Maxdist-enum` obtain a good diversity of solutions. (E) and (F) show the evolution of the distances in 30 independent runs.

This means that the most distant solutions are discovered at the beginning of the search and then there is less and less distance between new found solutions, something to expect given the reduced number of possible solutions in this example. Whether this small number of solutions (around 2% of the total number of equally valid solutions) is sufficient or not will depend on each particular case (for example, it can be enough to show an example of how extreme results can be in terms of different sets of reactions, but not enough to construct a good ensemble for the prediction of essential genes).

**Evaluation in realistic scenarios with an unknown number of optimal solutions.** In order to evaluate the diversity in a more biological setting, we randomly select different sets of highly expressed and lowly expressed enzymes of varying size in the Yeast 6 metabolic model

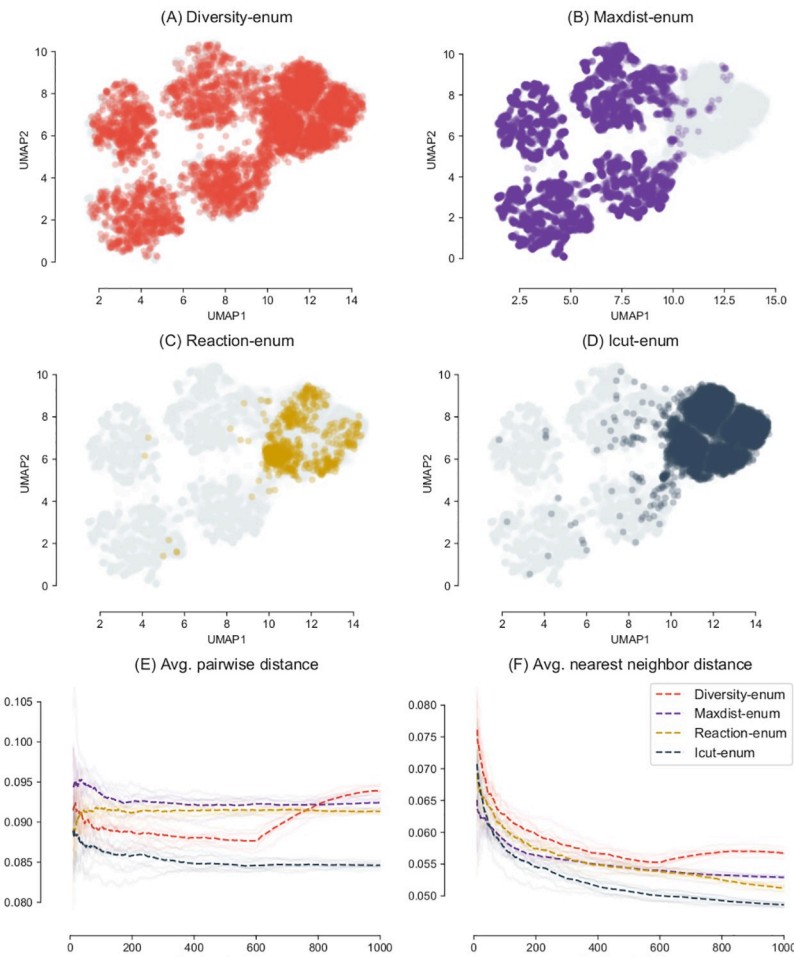

**Fig 6. Diversity of solutions in simulated problems using Yeast 6.** Enumeration of a maximum of 1,000 optimal metabolic networks on Yeast 6 [46] model, selecting a random set of reactions (120 $R_H$ and 120 $R_L$). Enumeration was repeated 10 times for each method, the average is represented with dashed lines in (E) and (F). The grey dots represent the union of all the solutions found by all the methods. `Diversity-enum` (A) shows a more homogeneous exploration of the space, exploring not only the distant solutions but also intermediate and close solutions.

[46] and then we enumerate a maximum of 1,000 optimal metabolic networks with the different methods.

Fig 6 shows the results of the enumeration of up to 1,000 optimal sub-networks from a randomly selected set of 120 genes highly expressed and 120 genes lowly expressed on Yeast 6. Enumeration of optimal solutions was repeated 10 times for each method. Since in this case the true set of possible optimal solutions is not known, grey dots represent the union of all discovered optimal networks for all the methods.

Again, a similar pattern of dispersion of the optimal solutions can be observed as with the DAG model. `Diversity-enum` (Fig 6A) obtains a set of solutions that look well spread across the space of enumerated solutions. The `Maxdist-enum` method misses most of the large set of similar solutions that are recovered by the other methods. Both the `Reaction-enum` and the `Icut-enum` method explore a similar and restricted region of the space, although `Icut-enum` can sample more densely from the same region.

Differences between the methods in this more realistic context are more obvious, and `Diversity-enum` performs comparatively better than the other methods. After `Diversity-enum` generates an initial set of around 600 solutions, both the average distance and the average nearest neighbor distance start to grow surpassing the other methods. A similar pattern can be observed for different random sets of selected genes (S1, S2 and S3 Figs). The rate at which this distance grows depends on several factors, including: the distance between the initial solutions, the space of possible solutions, and the parameters $d_s$, which controls the rate at which the distance of new solutions increases (S4 Fig).

## Prediction of essential genes using ensembles is highly dependent on the strategy used for enumeration

Next, we evaluate the predictive capabilities of the different methods for in-silico prediction of essential genes in the model organism *Saccharomyces cerevisiae*. We used gene expression measured from yeast in aerobic conditions [18, 26]. Genes were classified into expressed and not expressed using different combinations of thresholds on the quantiles of the distribution as it is commonly done in context-specific network reconstruction. For instance, a threshold of [0.25, 0.75] indicates that genes whose normalized expression value are below the quantile 0.25 are classified as lowly expressed, whereas those above the quantile 0.75 are highly expressed. Reactions were splitted into $R_H$ and $R_L$ sets using the `mapExpressionToReactions` method from the COBRA Toolbox [27].

Essential genes in Yeast 6 [46] were curated using most updated information from YDPM (http://www-deletion.stanford.edu/YDPM) database and the SGD (https://www.yeastgenome.org) project [47] (S1 File). Genes that are essential due to mechanisms not directly related to metabolism were excluded from the set, as they cannot be predicted using FBA. In total, 188 genes out of the 900 in Yeast 6 are considered to be essential under aerobic conditions.

A maximum of 2,000 optimal networks were enumerated for each combination of threshold and method, using a time limit of 8h per threshold/method, and 5 min. timeout for each MILP problem. The lower bound of the biomass reaction was constrained to carry a small positive flux, to ensure that all initial sub-networks will allow biomass production and therefore could be used to simulate the effects of gene knockout on the biomass production using FBA. In-silico predictions of essential genes were carried out using COBRA Toolbox v3.0.6 [27], classifying each gene as essential if the flux through the biomass reaction was below 1% after KO.

Essential genes were predicted for each optimal network within the set of the optimal networks obtained by each method and threshold, but also for the ensemble of networks, by taking the union of the predictions as shown in Fig 3. That is, if a gene is predicted as essential by a single optimal network from a set of optimal networks enumerated using a given method and threshold, then the gene is classified as essential by the ensemble. Thus, in total, we generated 16 ensembles per method, one for each threshold.

Table 1 shows the True Positives Rate (TPR, sensitivity) and False Positive Rate (FPR, 1-specificity) of these ensembles. `Diversity-enum` achieves the best TPR for all thresholds, with the best overall TPR of 0.7713 for the threshold [0.25, 0.90], which corresponds to the correct classification of 145 genes out of the 188 essential genes in the dataset. These results are followed by the `Reaction-enum` method, which achieves the same TPR as `Diversity-enum` in 8 out of 16 tests, with a slightly lower FPR in 6 out of those 8 tests. In contrast, `Max-dist-enum` and `Icut-enum` ensembles are not very well positioned in terms of TPR, although both methods achieve the lowest rates of false positives for some ensembles. Concretely, the `Icut-enum` method obtained the lowest FPR in 9 out of the 16 tests.

**Table 1. True Positive Rate (TPR) and False Positive Rate (FPR) of the ensembles for the prediction of essential genes in Yeast 6, for the different methods and thresholds.** Ensembles were generated by taking the union of the predictions of all enumerated networks per method and threshold.

| Threshold | Method | TPR | FPR | Threshold | Method | TPR | FPR |
|---|---|---|---|---|---|---|---|
| 0.10 0.90 | Diversity-enum | **0.7234** | 0.1264 | 0.20 0.90 | Diversity-enum | **0.7181** | 0.1194 |
| | Reaction-enum | 0.7181 | 0.1053 | | Reaction-enum | 0.7128 | 0.1025 |
| | Maxdist-enum | 0.4255 | **0.0730** | | Maxdist-enum | 0.4734 | 0.0969 |
| | Icut-enum | 0.4681 | 0.0815 | | Icut-enum | 0.4628 | **0.0576** |
| 0.10 0.85 | Diversity-enum | **0.6755** | 0.0871 | 0.20 0.85 | Diversity-enum | **0.6649** | 0.0927 |
| | Reaction-enum | 0.6649 | 0.0829 | | Reaction-enum | **0.6649** | 0.0843 |
| | Maxdist-enum | 0.4521 | 0.0674 | | Maxdist-enum | 0.5053 | 0.0702 |
| | Icut-enum | 0.3617 | **0.0604** | | Icut-enum | 0.4096 | **0.0365** |
| 0.10 0.80 | Diversity-enum | **0.7128** | 0.0688 | 0.20 0.80 | Diversity-enum | **0.6755** | 0.0520 |
| | Reaction-enum | **0.7128** | 0.0716 | | Reaction-enum | 0.6596 | 0.0716 |
| | Maxdist-enum | 0.4149 | **0.0379** | | Maxdist-enum | 0.4521 | **0.0337** |
| | Icut-enum | 0.4096 | 0.0534 | | Icut-enum | 0.3670 | 0.0562 |
| 0.10 0.75 | Diversity-enum | **0.6649** | 0.0843 | 0.20 0.75 | Diversity-enum | **0.6702** | 0.0590 |
| | Reaction-enum | **0.6649** | 0.0744 | | Reaction-enum | **0.6702** | **0.0520** |
| | Maxdist-enum | 0.4096 | **0.0548** | | Maxdist-enum | 0.4096 | 0.0632 |
| | Icut-enum | 0.3723 | **0.0548** | | Icut-enum | 0.3670 | 0.0534 |
| 0.15 0.90 | Diversity-enum | **0.7287** | 0.1053 | 0.25 0.90 | Diversity-enum | **0.7713** | 0.1334 |
| | Reaction-enum | 0.7234 | 0.1096 | | Reaction-enum | 0.7340 | 0.1194 |
| | Maxdist-enum | 0.4681 | **0.0758** | | Maxdist-enum | 0.5213 | 0.1067 |
| | Icut-enum | 0.4628 | 0.0983 | | Icut-enum | 0.4787 | **0.0913** |
| 0.15 0.85 | Diversity-enum | **0.7128** | 0.0829 | 0.25 0.85 | Diversity-enum | **0.6862** | 0.0885 |
| | Reaction-enum | **0.7128** | **0.0576** | | Reaction-enum | 0.6649 | 0.0843 |
| | Maxdist-enum | 0.5000 | 0.0927 | | Maxdist-enum | 0.4574 | 0.0730 |
| | Icut-enum | 0.3617 | **0.0576** | | Icut-enum | 0.4202 | **0.0590** |
| 0.15 0.80 | Diversity-enum | **0.7021** | 0.0885 | 0.25 0.80 | Diversity-enum | **0.6702** | 0.0871 |
| | Reaction-enum | **0.7021** | 0.0815 | | Reaction-enum | **0.6702** | 0.0576 |
| | Maxdist-enum | 0.3989 | **0.0323** | | Maxdist-enum | 0.4096 | 0.0590 |
| | Icut-enum | 0.3670 | 0.0548 | | Icut-enum | 0.4521 | **0.0534** |
| 0.15 0.75 | Diversity-enum | **0.6649** | 0.0506 | 0.25 0.75 | Diversity-enum | **0.6755** | 0.0843 |
| | Reaction-enum | **0.6649** | 0.0801 | | Reaction-enum | 0.6702 | 0.0801 |
| | Maxdist-enum | 0.4309 | **0.0534** | | Maxdist-enum | 0.4096 | 0.0604 |
| | Icut-enum | 0.4043 | 0.0562 | | Icut-enum | 0.4096 | **0.0548** |

Differences between ensembles can be better assessed by placing each ensemble in a ROC space (Fig 7), in which each point is an ensemble represented by its TPR and FPR. The upper part of the figure is dominated by `Diversity-enum` and the `Reaction-enum` method, whereas the `Maxdist-enum` and `Icut-enum` ensembles are characterized by a lower ratio of true and false positives.

One reason that explains these differences between the methods is the systematic generation of alternative solutions by testing every reaction in the model. If one reaction associated to a gene that is essential is not present in any of the set of optimal networks, the gene is not predicted to be essential. However, if there exist at least one optimal solution in which this reaction is present and essential, both `Reaction-enum` and `Diversity-enum` have more chances to detect it as they are going to test if there exist an optimal network with that reaction being active. `Maxdist-enum` and `Icut-enum` methods leave many of these solutions unexplored. `Diversity-enum`, in contrast, uses the `Reaction-enum` strategy to

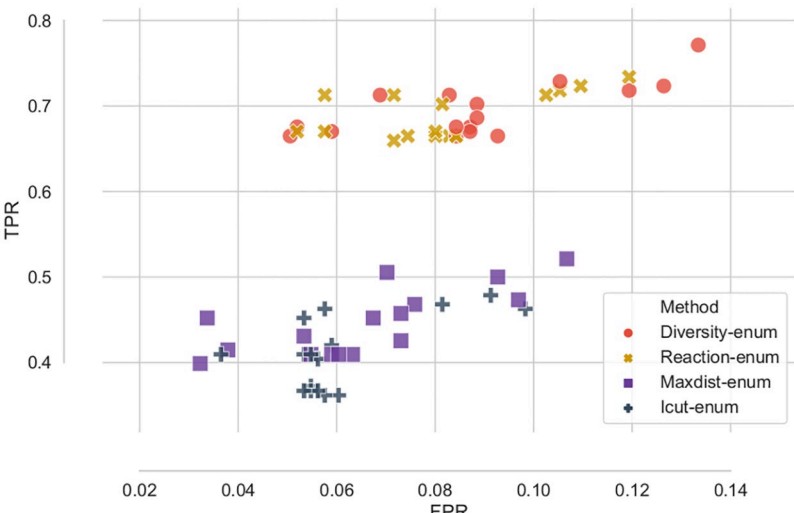

**Fig 7. Performance of each ensemble (TPR/FPR) for each method and threshold.** Each point represents the score (FPR, TPR) achieved by each ensemble built using a concrete threshold and enumeration method (data from Table 1).

have an initial set of solutions with variations in single reactions, from which it expands the search incrementally, increasing the chances of detecting even more essential genes.

Differences in TPR (S6 Fig) and FPR (S7 Fig) for the ensembles show that the individual networks generated by the different methods achieve a similar rate of true positives and false positives, and so the higher rates scored by the ensembles using `Diversity-enum` and `Reaction-enum` are driven by a more diverse set of predicted essential genes. That is, individual networks enumerated by these methods are able to correctly predict different sets of true essential genes, and so the union of those predictions include a more diverse set of detected essential genes. Concretely, the median TPR for the ensembles generated with `Diversity-enum` and `Reaction-enum` increase 142% with respect the median TPR of their individual networks, whereas the TPR of the ensembles built with `Maxdist-enum` and `Icut-enum` increase only 54% and 51% respectively.

In order to test whether the distance parameter $d_s$ has some strong impact on the results obtained with `Diversity-enum` method, we repeated the same experiment with parameter values $d_s = 0.990$ and $d_s = 0.999$ (S5 Fig). In both cases, the results obtained are very similar to these results obtained with the default parameter $d_s = 0.995$. This result suggest that most relevant solutions to the problem of prediction of essential genes are concentrated in the same region of the space of optimal solutions that is explored by both `Reaction-enum` and `Diversity-enum`. This space corresponds to the alternative solutions generated by modifying the constraints of single reactions in the networks (forcing the inclusion or knocking-out the reaction). Since the simulation of essential genes is based on simulating knockouts in the reactions associated with the genes, it is likely that most of the essential genes can be predicted in some of the optimal networks resulting from those variations in single reactions. However, there is still an advantage in using `Diversity-enum`, since it expands the initial set of solutions and is able to search for many more than the other technique is not capable of, increasing the probability of detecting more relevant reconstructions.

The computational time of each of the techniques is also different, although it depends on the size of the network and the number of variables (reactions associated with highly and lowly expressed genes). In general, `Reaction-enum` is the fastest method, while `Maxdist-enum`

is the slowest technique, since the optimization problem of looking for the farthest solution at every step has a higher computational cost (S1 Appendix).

## Diversity-enum detects more alternative hypothesis of the possible metabolic state of different human cancer cells

Next, we evaluate the ability to characterize the variability of predictions about which metabolic pathways are most active in different cancer cell lines. To do so, we reconstruct many optimal networks for each cancer cell line and we perform pathway enrichment on each network to see which pathways are more represented in the reconstructed networks than would be expected by chance. Given that there are multiple possible optimal reconstructions per cancer cell line, performing pathway enrichment on each optimal network will give different p-values for each pathway. This variability due to the method can have important implications. For example, by performing pathway enrichment on a single metabolic network, for a given significance level (e.g. $\alpha = 0.05$) we can detect that pathway A is enriched whereas pathway B is not. However, if we enumerate the space of optimal solutions, we can find an alternative solution in which pathway B is enriched but pathway A is not. Reporting pathway enrichment p-values of a single context-specific metabolic network without characterizing the variability should be in general avoided, as these values are misleading.

In order to test the variability in pathway enrichment scores due to the alternative set of optimal solutions, we used data from [19] for melanoma cells (cell line A375) and leukemia cells (HL60, K562, and KBM7 cell lines) and the human Recon 1 model [48]. Table 2 shows the enrichment results for the reconstructions using two different gene expression thresholds. Column *#Nets* shows the number of optimal networks that each method was able to enumerate (in a time limit of 8 hours). Column *#Enr.* shows the number of different enriched pathways (adjusted p-value < 0.05) that were detected by each method. It is important to remark that here, detecting more enriched pathways is better, since all the methods explore the same optimal solutions (all methods for enumeration maximize the same objective function and use the same experimental data). Detecting less enriched pathways means that there exist some other alternative metabolic networks that are enriched for other pathways but the enumeration method missed it, reporting that no enrichment was detected in any of the enumerated metabolic networks.

Overall, the method *Diversity-enum* is able to discover more alternative hypotheses about the pathways that are most active in each cell line, especially for the threshold [0.10, 0.90]. One of the reasons why there are more pathways that can be enriched is that, with this threshold,

**Table 2. Number of optimal networks (#Nets) in a time limit of 8 hours, and number of different enriched metabolic pathways (#Enr. i.e., pathways with p-value < 0.05 using the one-sided Fisher's exact test for over-representation, corrected for multiple hypothesis comparisons using the Benjamini-Hochberg procedure) for each cell line and gene threshold.**

| Cell line | Threshold | Diversity-enum | | Reaction-enum | | Icut-enum | | Maxdist-enum | |
|---|---|---|---|---|---|---|---|---|---|
| | | #Nets | #Enr. | #Nets | #Enr. | #Nets | #Enr. | #Nets | #Enr. |
| A375 | [0.10, 0.90] | 2933 | **31** | 2230 | 28 | 2804 | 23 | 3002 | 27 |
| A375 | [0.25, 0.75] | 1439 | **12** | 2278 | 11 | 892 | 11 | 1364 | 11 |
| HL60 | [0.10, 0.90] | 2855 | **28** | 2200 | **28** | 3004 | 23 | 3001 | 27 |
| HL60 | [0.25, 0.75] | 1450 | **13** | 2290 | **13** | 752 | 9 | 1223 | 12 |
| K562 | [0.10, 0.90] | 2934 | **29** | 2208 | 25 | 3006 | 25 | 1835 | 28 |
| K562 | [0.25, 0.75] | 1406 | 12 | 2283 | **16** | 532 | 11 | 1274 | 13 |
| KBM7 | [0.10, 0.90] | 2876 | **27** | 2150 | 25 | 2914 | 22 | 3000 | 26 |
| KBM7 | [0.25, 0.75] | 1793 | **13** | 2206 | **13** | 1384 | 12 | 1892 | 13 |

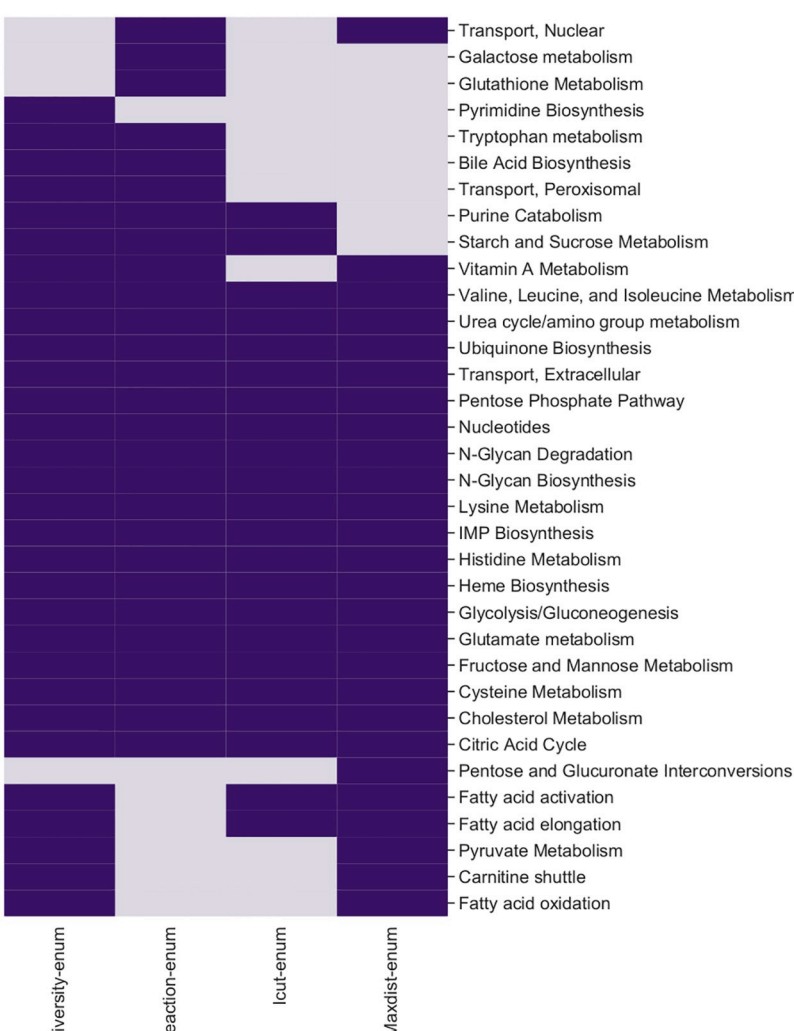

**Fig 8. Pathway enrichment results for the cell line A375 (human melanoma), threshold [0.10, 0.90].** Dark purple indicates that the method detected at least one optimal metabolic network for which the corresponding pathway was overrepresented (p-value < 0.05, B-H adjusted one-sided Fisher's exact test). Only pathways from Recon 1 that were enriched in some optimal solution (by any of the methods) are shown.

far fewer genes are classified as expressed and not expressed, and therefore many reactions of the metabolic network remain unscored (they may or may not be active without affecting the optimality of the solution). This makes it much more likely to find alternative sub-networks whose flux is consistent for the selected genes. In other words, the fewer genes that are identified as expressed or not expressed (less constraints), the more possible hypotheses about the metabolic state are consistent with the data. This is especially relevant in studies using proteomic or exometabolomic data, where the number of identified proteins or metabolites is lower than in gene expression assays.

Among some of the differences, `Diversity-enum` detected the metabolic pathways *Fatty acid activation*, *Fatty acid elongation*, *Fatty acid oxidation* and *Carnitine shuttle* enriched in the reconstructions for the A375 cell line both for the thresholds [0.10, 0.90] (Fig 8) and

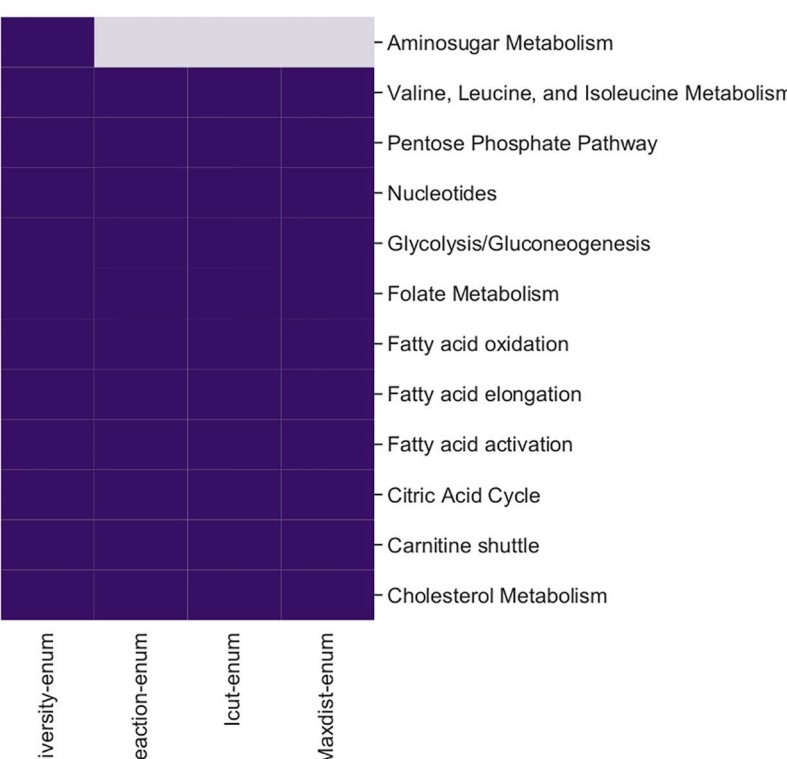

**Fig 9. Pathway enrichment results for the cell line A375 (human melanoma), threshold [0.25, 0.75].** Dark purple indicates that the method detected at least one optimal metabolic network for which the corresponding pathway was overrepresented (p-value < 0.05, B-H adjusted one-sided Fisher's exact test). Only pathways from Recon 1 that were enriched in some optimal solution (by any of the methods) are shown.

[0.25, 0.75] (Fig 9), whereas `Reaction-enum` detected them enriched only for the threshold [0.10, 0.90]. The single case where `Reaction-enum` discovered more existing alternative solutions with enrichment in other pathways not detected by `Diversity-enum` was for the cell *K562* for the threshold [0.25, 0.75], where was able to discover the alternative optimal solutions where pathways *Ubiquinone Biosynthesis, Cysteine Metabolism, Aminosugar Metabolism, and Urea cycle/amino group metabolism* were enriched in at least one of the optimal solutions enumerated with this method. `Icut-enum` was the strategy that obtained the worst results, not being able to find many of the optimal solutions with enrichment in other pathways that were discovered by the other methods. The `Maxdist-enum` method, although it detects in general less variation, finds in some cases enrichment in pathways that are not detected by any of the other techniques (e.g. Pentose and Glucoronate Interconversions in A375 for the threshold [0.10, 0.90], Fig 8). This could indicate that, although all solutions are equally valid, due to the topology of the network, the pattern of the distribution of highly expressed and low expressed genes across the network, and other factors such as the type of algorithms used by the solvers, there are certain types of solutions that are more frequently discovered than others, and therefore are more biased towards the discovery of this type of solutions. In these cases, the Diversity-enum $d_s$ parameter could be decreased to make the search spend more time exploring distant solutions.

We also analyzed how often each pathway was detected as enriched by any optimal metabolic network for each enumeration method (S1 Appendix). Among some of the possible causes that may affect this variability, the number of selected highly and lowly expressed has a clear impact in the results, for the reasons discussed before. Using a threshold of [0.10, 0.90], it can be seen that this variability is greater than for the threshold [0.25, 0.75], simply due to the fact that in the first case only 20% of the genes are used, and more hypothesis are consistent with the data. The enumeration of alternative reconstructions helps to characterize this variability and reduces the risk of incorrectly discarding hypotheses that are equally valid for the same reconstruction method and the same data.

## Discussion

Context-specific metabolic network reconstruction is a widely used approach to integrate different layers of experimental data into metabolic networks. This process allows to capture the metabolic sub-network that corresponds to the active part of the metabolism of an organism in a given condition. Using this reconstructed model, more advanced techniques such as Flux Balance Analysis (FBA), pathway enrichment, network visualization or gene essentiality prediction can be used to get an integrated view of the metabolic behavior.

One important limitation with this methodology is that context-specific metabolic network reconstruction is subject to significant variability due to the large number of optimal metabolic networks that can be reconstructed for the same experimental data, among other factors. This variability, which is commonly neglected, can contain relevant information and can offer alternative hypothesis of the metabolic state in terms of different combinations of reactions that are predicted to be active or inactive. Thus, the report of results using only a single optimal context-specific metabolic network can be highly biased and can overlook information relevant to the experiment. While this is an important issue, the analysis of the alternative set metabolic networks is a topic not well explored.

In this study we analyze the problem of enumeration of multiple optimal context-specific metabolic networks both from a theoretical and practical perspective. We show how it is common to have multiple different context-specific metabolic networks that optimally explain the same observed experimental data. The set of optimal solutions constitute different hypotheses of the metabolic state and therefore must be taken into account to reduce bias in the interpretation of results.

We propose four different methods for enumeration of context-specific reconstructions (`Reaction-enum`, `Icut-enum`, `Maxdist-enum` and `Diversity-enum`) that we developed and integrated in an unified open-source library called `DEXOM`. The first three methods are improvements of previous ideas that we have adapted and improved for the particular case of the enumeration of context-specific metabolic networks, whereas `Diversity-enum` is a novel method for enumeration of optimal solutions that maximizes incrementally the diversity.

We evaluate the methods focusing on two main aspects: 1) diversity of the optimal solutions obtained with each method, using two different distance metrics and UMAP plots to evaluate the spreading of the solutions; and 2) the biological relevance of alternative optimal solutions by assessing the predictive capabilities with real data. For this second aspect, we evaluate, on the one hand, the improvement in in-silico predictions of essential genes in *Saccharomyces cerevisiae* using ensembles of diverse metabolic network, and on the other hand, the detection of alternative enriched pathways in human cancer cells, as a way to measure the variability of different hypotheses about the metabolic state that are compatible with the experimental data.

In terms of distance metrics and the spread of solutions, both `Diversity-enum` and `Maxdist-enum` achieve good results, although `Diversity-enum` explores the solution space in a more homogeneous way than `Maxdist-enum`, which looks for more solutions in the extremes. `Reaction-enum` has a limited exploration capacity, focusing on similar solutions that represent a small part of the total solution space. `Icut-enum`, although capable of enumerating more solutions than `Reaction-enum`, does so in a much less diverse manner than `Diversity-enum` and `Maxdist-enum`, and sometimes even with less diversity than `Reaction-enum`, as reflected in the simulations using the Yeast 6 model [46].

With respect to predictive capabilities of essential genes using the Yeast 6 model, on an individual basis there are not large differences in terms of True Positive Rate (TPR) and False Positive Rate (FPR) between the individual optimal metabolic networks enumerated by each method. However, when the results are combined using ensembles of optimal metabolic networks, the TPR of the ensemble obtained with `Diversity-enum` increases by 140% compared to the median TPR of the individual networks, whereas ensembles generated with the methods that generate less diverse sets of solutions achieved only an increment of 50%. `Diversity-enum` was also the method with the best overall TPR of 0.7713, which corresponds to 145 out of 188 correctly classified essential genes, for a FPR of 0.1334 (95 false positives out of 712 non essential genes). These differences are explained by a more diverse set of essential genes captured by the individual optimal networks enumerated with `Diversity-enum`. This suggests that `Diversity-enum` allows to retrieve sub-networks that are more diverse in terms of metabolic pathways that can be used to reach the metabolic state that conforms to the gene expression data, and allows to explore a more diverse metabolic activity that is consistent with the same experimental data.

The technique `Reaction-enum` is also able to generate good ensembles, achieving a similar FPR and TPR as `Diversity-enum`, while techniques `Icut-enum` and `Maxdist-enum` obtain much worse results, regardless of the diversity or total number of solutions discovered. This might be explained in part by the fact that the relevant set of solutions for this problem is mostly confined to a small region in the space of optimal solutions, which corresponds to the space that both `Diversity-enum` and `Reaction-enum` are able to explore.

In terms of alternative hypothesis of the metabolic state of different human cancer cells, results obtained using pathway enrichment on the set of the optimal networks discovered by each method show that `Diversity-enum` is able to discover, in almost all cases, more diverse solutions in terms of networks that are enriched for other pathways alternative solutions. These results are again followed by `Reaction-enum`, `Maxdist-enum` and finally `Icut-enum`.

One important limitation of enumerating optimal solutions is the heavy computational cost involved in the search process. If the number of highly expressed genes and lowly expressed genes is very large, obtaining a single optimal metabolic network can be computational demanding or even not feasible in reasonable time, since obtaining an optimal solution involves solving a MILP problem, which is in general NP-Hard. In this context, enumerating multiple optimal solutions can be prohibitively expensive in some cases, especially with techniques like `Maxdist-enum` or `Diversity-enum`. One thing that can be done in these situations to alleviate the computational burden is to reduce the integer optimality tolerance of the solver to stop looking for better solutions once the solver has found a feasible integer solution proved to be, for example, within 1% of optimal.

Overall, this work provides different methods to explore the space of alternative context-specific metabolic network reconstructions, and an extensive comparison under different settings. We generated in total around 191,000 network reconstructions with simulated data,

around 329,000 reconstructions using microarray data from *Saccharomyces cerevisiae* and around 67,400 using RNA-seq data from different human cancer cell lines. Results of this evaluation show the importance of using an enumeration technique that finds a diverse set of solutions for different biological contexts. These results also provide important information for deciding which technique to use in each case. In general, `Diversity-enum` is the one that detects the most varied and relevant solutions for different biological contexts, followed by `Reaction-enum`, `Maxdist-enum` and `Icut-enum`. Our study also highlights that, given the variability of space of possible solutions that exists for a context-specific reconstruction problem, the analysis of a single solution, as is usually done, is not recommended, and downstream analysis made on a single metabolic network should be taken always with caution.

## Supporting information

**S1 Fig. Enumeration in Yeast 6 with 100 random $R_H$ and 100 random $R_L$ reactions.**
(TIF)

**S2 Fig. Enumeration in Yeast 6 with 80 random $R_H$ and 80 random $R_L$ reactions.**
(TIF)

**S3 Fig. Enumeration in Yeast 6 with 60 random $R_H$ and 60 random $R_L$ reactions.**
(TIF)

**S4 Fig. Example of the effect of the parameter $d_s$ used in the Diversity-enum method.** Values closer to 1 make the enumeration progress more slowly from closer to distant optimal solutions, discovering more proximate and intermediate solutions. When the value is lower (e.g. 0.990), the enumeration reaches the distant solutions more quickly, enumerating more solutions at the extremes.
(TIF)

**S5 Fig. Analysis of the effect of the parameter $d_s$ used in the Diversity-enum method for the prediction of essential genes.** Analysis was repeated with parameter values $d_s = 0.990$ and $d_s = 0.999$ instead of the default value ($d_s = 0.995$). The results show almost no variation in terms of the TPR and FPR of the ensembles.
(TIF)

**S6 Fig. Distribution of the True Positive Rates (TPR).** Results show 1) the variability in the predictions of true essential genes by individual networks, and 2) the result of the ensemble for each method. Although variation of results of the individual solutions enumerated with each method are similar, results of the ensemble greatly differ between methods. This indicates that although the individual networks predict a similar number of true positive essential genes, these sets present less overlapping in networks enumerated with `Diversity-enum` and `Reaction-enum`, and therefore the overall TPR of the ensembles generated by these methods is better.
(TIF)

**S7 Fig. Distribution of the False Positive Rates (FPR).** Results show, as for the TPR, an increase of the FPR of the ensembles, more pronounced for `Diversity-enum` and `Reaction-enum`, since there is always a trade-off between both measurements: increasing the predictions of true positives comes with a higher rate of false positives.
(TIF)

**S1 File. Essential genes in Yeast.** Dataset used for the evaluation of in silico prediction of essential genes in Yeast. We used annotations from SGD [47], YDP (https://www.yeastgenome.org/) and Yeast 6 [46] to classify each gene as essential or not under aerobic conditions. (XLSX)

**S1 Appendix. Supplementary information.** PDF including: 1) computational time of the different enumeration techniques; 2) distribution of predicted essential genes among methods; and 3) variability in the detection of enriched pathways between the different methods. (PDF)

## Author Contributions

**Conceptualization:** Pablo Rodríguez-Mier, Nathalie Poupin, Fabien Jourdan.

**Data curation:** Pablo Rodríguez-Mier.

**Formal analysis:** Pablo Rodríguez-Mier.

**Funding acquisition:** Laurent Le Cam, Fabien Jourdan.

**Investigation:** Pablo Rodríguez-Mier, Nathalie Poupin, Fabien Jourdan.

**Methodology:** Pablo Rodríguez-Mier, Nathalie Poupin, Fabien Jourdan.

**Project administration:** Laurent Le Cam, Fabien Jourdan.

**Resources:** Carlo de Blasio, Laurent Le Cam.

**Software:** Pablo Rodríguez-Mier, Nathalie Poupin.

**Supervision:** Nathalie Poupin, Fabien Jourdan.

**Validation:** Pablo Rodríguez-Mier, Nathalie Poupin, Carlo de Blasio, Laurent Le Cam.

**Visualization:** Pablo Rodríguez-Mier.

**Writing – original draft:** Pablo Rodríguez-Mier.

**Writing – review & editing:** Pablo Rodríguez-Mier, Nathalie Poupin, Carlo de Blasio, Laurent Le Cam, Fabien Jourdan.

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
