## [Decision Letter · Decision Letter 0]

10 Sep 2020

Dear Dr. Rodríguez Mier,

Thank you very much for submitting your manuscript "DEXOM: Diversity-based enumeration of optimal context-specific metabolic networks" for consideration at PLOS Computational Biology.

As with all papers reviewed by the journal, your manuscript was reviewed by members of the editorial board and by several independent reviewers. In light of the reviews (below this email), we would like to invite the resubmission of a significantly-revised version that takes into account the reviewers' comments. In the revisions, as pointed out by the reviewers, please make particularly sure to provide more details on the insights which can be achieved through using an ensemble of metabolic models and expand the applications to another organism with a larger metabolic network (e.g. human) to demonstrate the broad applicability of your method.

We cannot make any decision about publication until we have seen the revised manuscript and your response to the reviewers' comments. Your revised manuscript is also likely to be sent to reviewers for further evaluation.

Sincerely,

Christoph Kaleta

Associate Editor

PLOS Computational Biology

Daniel Beard

Deputy Editor

PLOS Computational Biology

Reviewer's Responses to Questions

**Comments to the Authors:**

Reviewer #1: Reproducibility report has been uploaded as an attachment.

Reviewer #2: Overview:

In the manuscript, "DEXOM: Diversity-based enumeration of optimal context-specific metabolic networks", the authors describe a novel method that create a diversity of context-specific metabolic networks. The issue of alternative metabolic networks not being captured by various different context-specific extraction methods has been noted in the past. Previous papers, for e.g. Robaina-Estevez S et al., 2017, pointed out the effects of alternate optimal context-specific metabolic networks. However, like that paper, the author's results also appear to target iMAT and fastCORE. While I think alternate optimal context-specific metabolic network do contain important results, I haven't seen anyone explicitly show that improvements in context-specific models can be achieved by enumerating alternate optimal states. To this end, the authors have tested alternate optimal context-specific metabolic networks to show that - (a) only small improvements could be achieved over Rxn-enum, and (b) large improvements could be achieved over Maxdist and Integer-cut. Different algorithms are, in a way, different interpretations of the organization of context-specific network; thus, it should depend on the enumeration algorithms. Further, the premise misses the importance of thresholding as an input to the algorithm. Recently, many different thresholding methods have been proposed and reviewed which maybe able to perform better and may highlight power of DEXOM. DEXOM can be a promising method since it did The authors have tried to do some benchmarking but it also requires more testing to fully evaluate the power of DEXOM. How much can I learn more about my data/organism/biology by using DEXOM rather than other methods? Based on this, I recommend major revisions.

MINOR CONCERNS:

1. Please provide a line numbered version, so reviewers can point out the location of reference easily.

2. How prevalent is the usage of DAG in understanding metabolic networks. Perhaps authors could better highlight and justify their use of DAGs in Introduction.

3. In regards to TPR and FPR, the authors chose to report the ratios. Could they, please, also show the actual numbers? I am curious how they get influenced from difference between low to high thresholds.

4. Are Table 1, Figure 7, and Figure 8 coming from the same set of simulations? If yes, could authors please either concise some of this information into a single figure or put some of it in supplementary information?

5. Could the authors, please, provide the link to the method within the paper somewhere as well.

MAJOR CONCERNS:

1. I want to make sure that by optimal conext-specific model they mean, models which maximizes the addition of active and removal of inactive reactions with experimental evidence. Doesn't this definition assume that the output of the thresholding is accurate? Besides global thresholding used in this paper, recent studies have identified better thresholding approaches that generate more accurate models, also some thresholding approaches have enforced biological significance. Have the authors looked at some of these thresholding approaches to see if DEXOM can do better with better thresholds?

2. A lot of the work on how to measure diversity has been done, i.e. the diversity between models has often been reported using Jaccard similarity (see Fastcore line of papers and papers evaluating context-pecific extraction method). The behavior of this distance metric with reaction active with experimental evidence and those of the models has been shown in these papers. The authors can choose to use Hamming distance, but it becomes difficult for the reader to compare results in those papers with those reported by the authors. Could the authors please provide some justification for their choice to use Hamming distance? Alternative authors could also show that these results could be reproduced using Jaccard similarity.

3. Robaina-Estevez et al. 2017 found that CORDA, a methods that enforced metabolic functionality rather than the reactions performed better. Does this statement still hold true with DEXOM using gene essentiality as metric?

4. This is tied to point 4, how does DEXOM do if it is coupled with metabolic functionality.

5. For improving these types of context-specific models, two major ways have been targeted: (1) extraction method, and (2) thresholding. The authors are dealing with a novel extraction method. As a way to point out if this is the way forward, can the authors compare their TPR/FPR with those achieved from changing the thresholding method.

6. Different algorithms are enforcing different interepretations of how context-specific metabolic networks are organized. Whether the interpretation is true for only yeast, single-celled organisms, eukaryotes, or fungi, or other organisms too. Yeast is a single cell organism which can be grown in the lab relatively easily compared to mammalian cells. Extensive comparison data and models are available if DEXOM was applied to human tissues, mouse, and/or cancer cell lines. Could the authors show that DEXOM can be extrapolated to other organisms and/or datasets?

Reviewer #3: Rodríguez-Mier et al. propose DEXOM, a diversity based enumeration method of optimal context-specific metabolic models. In essence the authors describe advantages and disadvantages of three alternatives for enumerating alternative solutions of optimal context-specific metabolic models.

The authors show and describe a thorough understanding of the theoretical aspects of their work.

DEXOM takes into consideration the advantages of three alternative MILP methods (Rxn-enum, integer-cut, maxdist) and computes increasingly distant (diverse) optimal alternative solutions based on an initial sampling set (computed with Rxn-enum). Diversity and comparison to other methods is quantified by two metrics, hamming and nearest neighbor distance, which are used throughout the manuscript for evaluation across different methods. The capability of DEXOM to predict gene essentiality based on published data and a model for S. cerevisae are demonstrated.

Though the authors identified and addressed a very important aspect of constraint based modeling, I see major drawbacks that prevent the laid out DEXOM algorithm to be published in PLOS COMP BIO.

Although the diversity seems to be sampled best by DEXOM as visually shown by UMAP based dimension reduction, the advantage of the improved metrics seems to be not always the case as shown in panels E and F of Fig. 5 and S1-S3 and depends to a large quantity on the sampled networks (cf. e.g. S3 against S2 and S1). Hence, it appears that it cannot be guaranteed that DEXOM outperforms Maxdist with respect to the given metrics. In addition, as shown in Table 1 and Figure 7 and 8, the advantage of using DEXOM over Rxn-enum for finding essential genes in a given network is not substantial. Specifically the increased TPR is often achieved at the price of an increased FPR, while the differences in TPR and FPR for Rxn-enum and DEXOM are generally very low. This seems to render Rxn-enum the better algorithm for identifying gene essentiality due to its low runtime (compared in Fig. 9), despite providing only a restricted and not diverse set of optimal networks.

In summary, the advantages of using and applying DEXOM are not obvious and the authors need to justify why DEXOM should be used instead of Rxn-enum (for finding essential genes) or Maxdist (for computing a diverse set alternative optimal networks for further downstream analysis like pathway enrichment).

With this several further major drawbacks are present:

- No usable DEXOM algorithm is provided. This work would greatly benefit from having a ready made software code available that based on any given SBML model allows to compute a diverse set of alternative optimal metabolic models.

- Only one model and one dataset on yeast are used for demonstrating and comparing the performance of the proposed algorithm and this only for gene essentiality analysis. Although mentioned in the introduction as potential use case the authors did not include any model analysis to investigate diseases in e.g. human, presumably because human reconstructions like Recon3D possess a large number of reactions, which make it computationally demanding to compute optimal solutions. Nevertheless, it remains unknown how DEXOM performs on other data. At least one further data set analysis, potentially with a different aim than identifying gene essentiality (e.g. enriched function) of another organism would greatly improve the trust in DEXOM, if it can be shown that it outperforms competing methods.

- The authors do not discuss the virtually equal quality of Rxn-enum vs DEXOM for gene essentiality analysis. Furthermore, TPR and FPR are not given for the alternative sets presented in S1-S3, which show much different hamming and nearest neighbor values, which presumably influence achievable TPR and FPR values. Code and solutions are not given to allow recapitulating the analysis.

- Throughout the manuscript preciseness is missing. Parameters and functions of the presented algorithm are not described (e.g. exp(a,b) = a^b is not explained). Figures are not described well (subpanel description are not given in captions and are most often not referenced in the main text). Another example is the definition and use of thresholds used in table 1, which need to be briefly introduced in the main text (e.g. at page 20).

- A major rewrite is recommended, as much of the methods section is recapitulating methods that are published elsewhere. Instead the focus should be on the DEXOM description, while the extensive text body on alternate methods (essentially page 4-13) leading towards DEXOM should be shortened dramatically in the main text. The detailed description would be suitable for a supplementary text that can be referenced in the methods.

Minor

==

- page 3: 10.1016/j.cels.2017.01.010 and https://doi.org/10.1371/journal.pcbi.1003580 are two references that should be mentioned here as they show performances of methods tailored towards computing context specific metabolic models

- page 8: "... extensively exploited in commercial solvers such as IBM CPLEX and

Gurobi" - both provide academic free academic licenses, which would be great to mention

- page 9: Here, a short paragraph on model optimisation by gap filling, detection of orphan reactions or EGCs leading to the fact that this does solve only parts of the problem of network artefacts would be suitable.

- page 14: "It starts computing an initial set of solutions using the Rxn-enum method avoiding duplicated solutions. This guarantees that single variations of reactions across all pathways are explored." - This is confusing, as the initial set of solutions will not necessarily include variations across pathways, unless the initial set size is sufficiently high.

- page 14: "Using a ds value close to one (e.g. ds = 0.99), the search concentrates at the beginning with more probability in the close vicinity of the selected solution." - Why should it be desirable to start with similar solutions as most diverse solutions are the goal? Of course the formulation allows to start with more distant solutions. It should be made clear here, why similar solutions might be desirable to be computed first.

- page 15: "For example, in Saccharomyces Cerevisiae, gene ARG2, which encodes glutamate N-acetyltransferase —a mitochondrial enzyme that catalyzes the first step in the biosynthesis of the arginine— is essential only in the

absence of arginine in the medium." - It should be made clear that this is might be a general theme in arginine metabolism, not just in S cerevisae (lowercase for cerevisiae)

- page 15: COBRA toolbox reference should be cited here and elsewhere. Also Yeast 6 is not referenced everywhere (e.g. page 16)

- page 17: "The grey points correspond to the 1,024 optimal solutions of the ground truth." - This is very unclear at this point. Please improve clarity of description of the figures here and elsewhere.

- page 17: "The Maxdist method shows at the beginning of the search the largest distance, since the solutions are generated by finding extreme differences. After an initial set of 25 optimal solutions, the average distance stops increasing. This is something to expect since the most distant solutions are usually discovered at the beginning of the search." - The question arises, whether having the first most diverse networks with Maxdist is sufficient and fast enough as both hamming and nearest neighbor distance are high. This should be analysed, mentioned and discussed.

- page 19: "After DEXOM generates an initial set of around 600 solutions, both the average distance and the average nearest neighbor distance start to grow surpassing the other methods." - It is surprising that there is not a gradual, but sudden improvement by DEXOM (also in Fig. 4 for the DAG model). The authors should explain or at least hypothesize on why this is happening abruptly and not monotoically increasing for both hamming and neirest neighboor distance. Regarding particularly investigation in the supplement it appears that for low numbers of sample networks, Maxdist is the better method in terms metric performances. Again, the authors should discuss this point.

- page 20: Since [38] lists gene expression under low oxygen levels (at most 2.8%), the authors should discuss briefly, whether there is a difference to be expected for higher oxygen levels and how this might affect gene essentiality.

- page 20: "Thus, in total, we generated 16 ensembles per method, one for each threshold." - Thresholds are not defined.

- Table 1 would be much easier to investigate as bar plot with four bars (4 methods) for each threshold configuration

- methods and results/discussion should be clearly separated (e.g. machine configuration mentioned on page 24 is Methods content)

**Have all data underlying the figures and results presented in the manuscript been provided?**

Reviewer #1: Yes

Reviewer #2: Yes

Reviewer #3: **No: **Computed models and essential genes are missing that are the basis for given manuscript and supplement figures and table 1.

PLOS authors have the option to publish the peer review history of their article (what does this mean?). If published, this will include your full peer review and any attached files.

Reviewer #1: No

Reviewer #2: No

Reviewer #3: **Yes: **Sascha Schäuble
---

## [Decision Letter · Decision Letter 1]

3 Dec 2020

Dear Dr. Rodríguez Mier,

Thank you very much for submitting your manuscript "DEXOM: Diversity-based enumeration of optimal context-specific metabolic networks" for consideration at PLOS Computational Biology. As with all papers reviewed by the journal, your manuscript was reviewed by members of the editorial board and by several independent reviewers. The reviewers appreciated the attention to an important topic. Based on the reviews, we are likely to accept this manuscript for publication, providing that you modify the manuscript according to the review recommendations. Please pay particular attention to the comments of reviewer 2 concerning the motivation to include all four methods in the work and the reasoning to include similarly performing methods with large differences in required runtime.

Sincerely,

Christoph Kaleta

Associate Editor

PLOS Computational Biology

Daniel Beard

Deputy Editor

PLOS Computational Biology

[LINK]

Reviewer's Responses to Questions

**Comments to the Authors:**

Reviewer #1: Reproducibility report has been uploaded as an attachment.

Reviewer #2: OVERVIEW:

In the manuscript, "DEXOM: Diversity-based enumeration of optimal context-specific metabolic networks", the authors describe a novel method that create a diversity of context-specific metabolic networks. It seems that enumerating multiple context-specific networks is valuable and is known for sometime as authors have pointed out. It is more clear in this version of the manuscript that authors are trying to bring everything under one umbrella called DEXOM and subtly claiming that diversity-enum is more accurate. My first major concern is that Diversity-enum and Rxn-enum seem to have the same accuracy. My second major concern is that current analysis doesn't say anything about networks uniquely captured by Diversity-enum, Rxn-enum, Icut-enum, or Maxdist. What is the problem that authors are solving - time taken to generate the ensemble, diversity in the ensemble, accuracy of the ensemble, or simply a toolbox? Based on the current analysis shown, I am still not clear if additional diversity, i.e. Diversity-enum, was useful in either improving calculation time or accuracy. It is also not clear why a modeler should care for all the 4 approaches that authors have described here, when clearly only Rxn-enum and Diversity-enum are in anyway accurate. Given that analysis presented here needs more fleshing out, I would recommend a major revision. Please see details below.

MAJOR CONCERNS:

1. Why is there a need for 4 different approaches which need to be under one umbrella? Why can't just the best method prevail? Are there any advantages of Icut-enum and Maxdist-enum? These two have very poor accuracy and why do they need to be included in the paper or in the toolbox. Can the authors show, systemically, cases where enumeration using one method is better than other. The authors can do this by finding which essential genes are captured differentially among networks calculated using different methods. I think this is an important benchmarking step for the authors to justify the need of multiple enumeration methods within this toolbox for context-specific networks.

2. To me it appears Rxn-enum and Diversity-enum have similar accuracy. Diversity-enum and Maxdist give the most diversity within the models and they both take a long time to run. So authors generated a method which produces a diverse ensemble but it takes a long time. However, this long time is not producing better results than Rxn-enum, seems pretty similar to me. I would use Rxn-enum because it goes really fast and produces that same accuracy as Diversity-enum. I am not sure if Diversity-enum produces valuable diversity within models. Unless authors can systematically show that extra context-specific metabolic networks that are being differentially captured by Diversity-enum but not by Rxn-enum are valuable. Authors can simply see which of the networks captured uniquely by Diversity-enum captured gene essentiality not explained by Rxn-enum.

3. If Diversity-enum is better, could the authors just do a statistical test for difference in distributions of TPR from different methods? These are basically p-values from comparing TPR distributions between pairs of methods.

Reviewer #3: The authors underwent substantial labor to improve their manuscript.

The focus of the paper has been improved and shifted away from focusing on the new DEXOM formulation itself (now termed Diversity-enum) to presenting a suite of readily usable algorithms for enumerating alternative optimal metabolic network solutions. These include Diversity-enum as one of four methods for which no COBRA compatible toolbox extension was available so far. The precision in phrasing has been improved throughout the manuscript. More importantly the authors succeeded in adding another use-case, by adding an enrichment analysis of four human cancer cell lines. The authors showed that a set of differing context specific metabolic models with equal optimality can possess a diverse set of enriched metabolic pathways. This outcome is a valuable addition to the manuscript, but misses one aspect to be complete (see below).

The authors show that equally probable context specific networks can possess a diverse set of enriched functions. In consequence pathway enrichment analysis needs to be treated with care. The authors do not show, however, how frequent any of these pathways occur, e.g. is there a single pathway that is enriched in most, if not in all, sets of networks? This way enriched pathways occurring only a few times, might be considered artifacts of the respective networks, yet pathways that are enriched in most networks hint towards metabolic function that translated into the context specific metabolic networks and thus would be of value. Here, it would also be of interest how the frequency of enriched pathways differs after applying the four different methods. Such an analysis should be easy to add, would improve the provided insights into enriched pathway analysis and complete the revision.

**Have all data underlying the figures and results presented in the manuscript been provided?**

Reviewer #1: None

Reviewer #2: Yes

Reviewer #3: Yes

PLOS authors have the option to publish the peer review history of their article (what does this mean?). If published, this will include your full peer review and any attached files.

Reviewer #1: **Yes: **Anand K. Rampadarath

Reviewer #2: No

Reviewer #3: **Yes: **Sascha Schäuble
---

## [Decision Letter · Decision Letter 2]

21 Jan 2021

Dear Dr. Rodríguez Mier,

We are pleased to inform you that your manuscript 'DEXOM: Diversity-based enumeration of optimal context-specific metabolic networks' has been provisionally accepted for publication in PLOS Computational Biology. 

Before your manuscript can be formally accepted you will need to complete some formatting changes, which you will receive in a follow up email as well as the remaining minor point raised by reviewer 3. A member of our team will be in touch with a set of requests.

Best regards,

Christoph Kaleta

Associate Editor

PLOS Computational Biology

Daniel Beard

Deputy Editor

PLOS Computational Biology

Reviewer's Responses to Questions

**Comments to the Authors:**

Reviewer #2: The authors have satisfactorily addressed all of my concerns.

Reviewer #3: The authors have followed up on my point raised and added a comprehensible analysis of the frequency of enriched pathways in the S1 appendix file.

These results, however, were not mentioned or referenced in the main text body.

The supporting information description was also not updated.

I kindly ask the authors to connect their frequency analysis of enriched pathways to the main text (presumably section "Diversity-enum detects more alternative hypothesis of the possible metabolic state of different human cancer cells") by adding e.g. a brief summary of their observation and linking the appendix file accordingly.

**Have all data underlying the figures and results presented in the manuscript been provided?**

Reviewer #2: Yes

Reviewer #3: Yes

PLOS authors have the option to publish the peer review history of their article (what does this mean?). If published, this will include your full peer review and any attached files.

Reviewer #2: No

Reviewer #3: **Yes: **Sascha Schäuble

---

## [Editor Report · Acceptance letter]

5 Feb 2021

PCOMPBIOL-D-20-01343R2 

DEXOM: Diversity-based enumeration of optimal context-specific metabolic networks

Dear Dr Rodríguez Mier,

I am pleased to inform you that your manuscript has been formally accepted for publication in PLOS Computational Biology. Your manuscript is now with our production department and you will be notified of the publication date in due course.

With kind regards,

Alice Ellingham
